# Early impact of agropastoral activities and climate on the littoral landscape of Corsica since mid-Holocene

**Marc-Antoine Vella**[1,2,3¤]*, **Valérie Andrieu-Ponel**[4], **Joseph Cesari**[5], **Franck Leandri**[5], **Kewin Pêche-Quilichini**[6], **Maurice Reille**[4], **Yoann Poher**[4], **François Demory**[1], **Doriane Delanghe**[1], **Matthieu Ghilardi**[1], **Marie-Madeleine Ottaviani-Spella**[2]

**1** UMR 7330 CEREGE, Aix Marseille University, Collège de France, IRD, INRA, CNRS, Aix-en-Provence, France, **2** UMR 6134 SPE, Corsica University, CNRS, Corte, France, **3** UMR 7619 METIS Sorbonne University, CNRS, EPHE, Paris, France, **4** Aix-Marseille University, Avignon University, IMBE, IRD, CNRS, Aix-en-Provence, France, **5** DRAC Corse, MMSH, LAMPEA, UMR 7269 CNRS, Ajaccio, France, **6** UMR 5140 ASM, Montpellier University, INRAP Méditerranée, CNRS, Montpellier, France

¤ Current address: UMR 7619 METIS Sorbonne University, CNRS, EPHE, Paris, France
* mav.vella@gmail.com

**Data Availability Statement:** All relevant data are within the manuscript and its Supporting Information files.

## Abstract

A multidisciplinary study (geomorphology, sedimentology and palynology) shows that the landscapes of the southwest coast of Corsica have been deeply modified by humans and the climate since 3000 BC. Significant and rapid landscape transformations are recorded between the Chalcolithic and the Middle Bronze Ages (3000–1300 BC). Several major (2.2 ka BC, 1.2 ka BC) and local (3000 BC) detrital events affected the Taravo Lower Valley in relation to global climatic changes and anthropic activities. The vegetation dynamics since 3000 BC show alternating phases of agriculture and abandonment until the complete disappearance of the original forest populations in the vicinity of the Canniccia Marshes. An early phase of *Olea* cultivation is recorded between 2900 and 2300 BC. Plant macro-remains indicate that cereals, vine and many species of Fabaceae were cultivated in the nearby of the archaeological sites during the middle to the late Chalcolithic Age. The event of 2.2 ka BC corresponds to an abandonment phase in the lower Taravo Valley. Pastoralism dominated agricultural activities between 2200 and 1700 BC. During Roman times, agriculture is characterized by olive and vine cultivation. A new peak of pastoralism and the cultivation of *Castanea* are noted during invasion times (500 to 1000 AD), showing that invasions didn't disturb agricultural activities in the Taravo Valley. During the Pisa Period (end of the 9th C. to then end of 13th C. AD), pastoralism declined and vine and cereals were cultivated in the very nearby of the Canniccia Marshes. During the Genoa Period upwards (end of the 13th C. to 1769 AD), a decline in agriculture and a recrudescence of the forest (maquis and pine) are recorded, leading to the settlement of a present-day vegetal landscape dominated by an *Erica arborea* maquis.

**Funding:** This work was supported by Ministry of Culture, DRAC/SRA Corse, ARTEMIS Program. MISTRALS/Archeomed/INEE. FIR (Fonds Incitatifs de Recherche, Université de Provence). The funders had no role in study design, data collection and analysis, decision to publish, or preparation of the manuscript.

**Competing interests:** The authors have declared that no competing interests exist.

## Introduction

Paleoenvironmental studies of Mediterranean Valleys have been widely developed since the late 1960s [1–10]. They confirm the importance of fluvial and lake environments, in order to better understand Holocene landscape changes [9, 11–17]. They also attest to the presence of periods of global rapid climate change (RCC) observed during the recent Holocene Epoch (2.2 ka BC, 1.2 ka BC, Little Ice Age (LIA, 1450–1850 AD) [9, 17–20]. These RCCs could have directly modified yields and agricultural practices, leading to major societal reorganizations, especially during the Bronze Age. These studies also demonstrated an increasing anthropogenic impact on Mediterranean ecosystems since the Neolithic Age [21].

However, few works exist about the Mediterranean islands on Holocene alluvial paleoenvironments. Most of actual references are largely limited to the Eastern Mediterranean where major civilizations arose (Minoan, Mycenian and Greeks). Cyprus [22–27], Euboea [28–32], and the north coast of Greece [20, 33] represent the most studied regions. Concerning the central Mediterranean island, several studies restitute the evolution of Sicilian lakes levels during the late Holocene [34–38]. A smaller number of studies treat the paleogeographic evolution of the littoral valleys in Malta [36, 39]. Finally, significant results for the Western Mediterranean were obtained in Sardinia [40] and Corsica [41–46] (Fig 1A).

In contrast, topics related to vegetation history and human impact on Mediterranean island ecosystems are more developed [47]. Pollen studies undertaken in Sicily ([35, 48–49], Crete

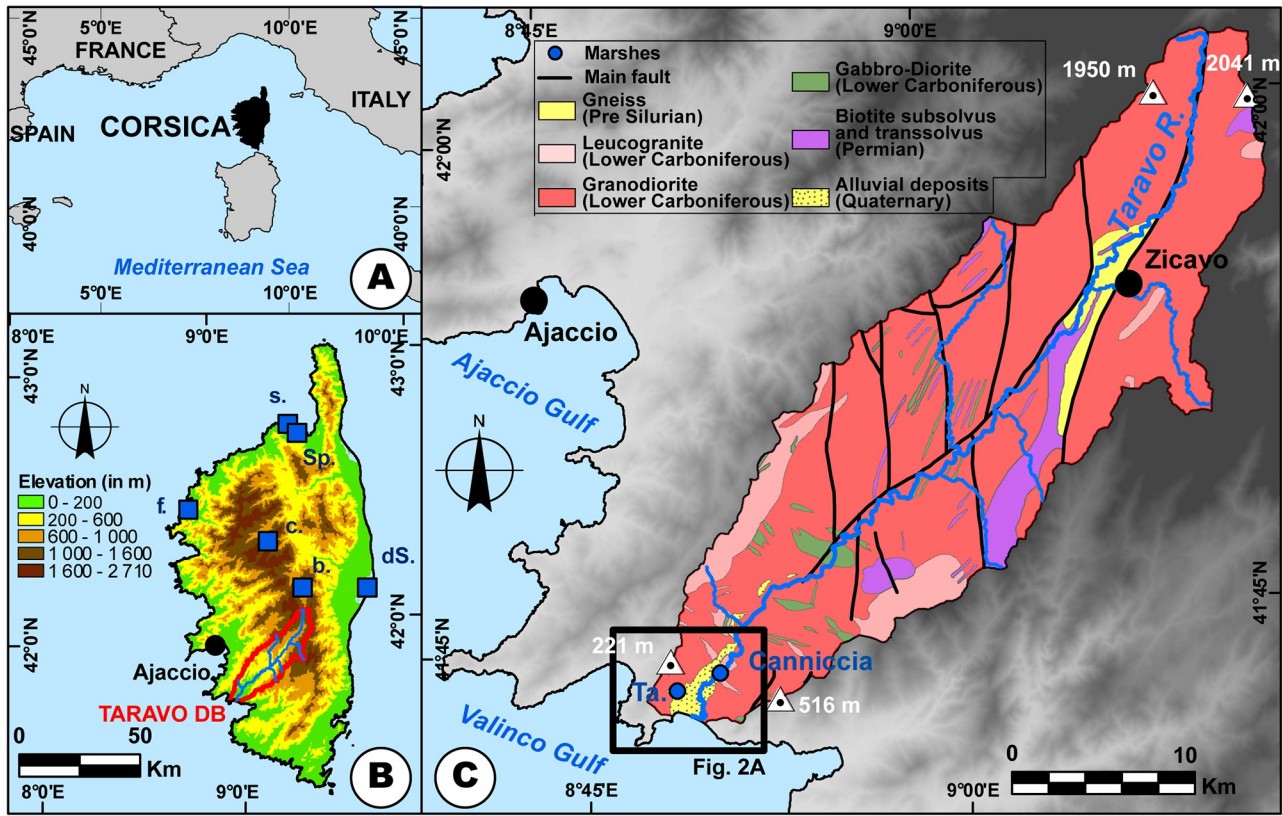

**Fig 1. Location map of the study area.** The data for the DEM (SRTM 3) was acquired from the USGS Earth Explorer Map Viewer (public domain). The data for the geology was acquired from the BRGM (Bureau de Recherches Géologiques et Minières) Map Viewer (public domain), modified. **A: Corsica within the western Mediterranean area. B: Topographic map of Corsica.** Reference site: b. Bastani, c. Creno, dS, Étang del Sale, f. Fango, s. Saleccia, Sp. Spizicciu. **C: Location and geology of the Taravo drainage basin**.

**Table 1. Geographic coordinates of the boreholes.**

| Borehole | Elevation above mean sea-level (in m) | Easting (in m) UTM WGS84 32N | Northing (in m) UTM WGS84 32N | Depth (in m) |
|---|---|---|---|---|
| CAN 1 | 9 | 8˚ 50' 54.3" | 41˚ 43' 39.5" | 5,8 |
| CAN 2 | 9 | 8˚ 50' 53.2" | 41˚ 43' 34.6" | 8,5 |
| CAN 3 | 9 | 8˚ 50' 57.3" | 41˚ 43' 39.1" | 3,2 |
| CAN REILLE | 9 | 8˚ 50' 56" | 41˚ 43' 35.7" | 15,8 |

Coordinates expressed in WGS 84 geodetic system

[50–51], Malta [36, 52–53] and Corsica [54–57] (Fig 1A) helped in reconstructing vegetation history since the end of the Last Glacial Maximum (LGM). Combined with micro-charcoal analysis in Corsica [58–63], these studies related human activities, such as fire signals, to vegetation change since the early Neolithic Age. This influence resulted in the modification of the vegetation composition/morphology, domestication and introduction of new species. However, from paleoenvironmental, geomorphological and geoarchaeological points of view, the alluvial plains of Corsica are relatively little studied (Fig 1A). Previous research on Quaternary, and Holocene sedimentary dynamics, in particular, remains fairly general [64–65] and makes no reference to possible impacts on population dynamics or relations to vegetation history.

This article focuses on the Taravo Valley where, since the 1950s, several archaeological research programs allowed to outline the main stages of cultural evolutions since the Mesolithic Age. However, they required more paleoenvironmental analysis especially for periods of major archaeological occupation of the island as few cross-combined approaches were realized on main archaeological sites. For the present study, three cores were drilled in the Canniccia Marshes where a previous deep core (CAN REILLE) was realized in 1998 [66] for pollen analyses (S1 Fig, Table 1, Fig 2A and 2B). In this article, only the first 10 m of CAN REILLE are presented.

CAN 1 (5.8 m), CAN 2 (8.5 m) and CAN 3 (3.2 m) were drilled in 2008 and studied for sediment properties. Preliminary results from recent geoarchaeological studies [42, 45–46] highlighted several detrital events in the Canniccia Marshes, just beneath the archaeological site of I Calanchi/Sapar'Alta. For the purpose of this article, the chronostratigraphic sequence has been reused and we have added the data from a new coring (CAN 3). The unpublished pollen data of CAN REILLE shed new light on the cross-impact of civilizations and climate on landscapes.

## General background

### Geological and geomorphological settings of the Taravo River

The Taravo River (Fig 1B), the third largest drainage basin of Corsica (487 km$^2$), is ca. 66 km long [67, 68]. Located in Southwest Corsica and oriented roughly NE-SW (Fig 1), its estuary is situated at ca. 25 km south of Ajaccio. Drainage basin of the Taravo is mainly represented by Paleozoic to Mesozoic granitic formations (granodiorite, monzonitic granite and leucocratic granite [69–72] (Fig 1C). However, metamorphic rocks (gneiss and migmatic rocks) and rhyolite have been identified in the center of the watershed. In the lower valley, ultrabasic rocks (gabbro-diorite) are present [72–74] (Fig 1C). Messinian regression and intense fault activity during the Oligocene-Miocene Epochs (between 20.5 and 15 Ma) [75–76] created the main orientation and morphology of the valley. Pliocene sedimentary formations are present in the form of relictual deposits (clay, conglomerate) on the slopes of the lower valley [71] (Fig 1C). Sedimentary formations of the Taravo alluvial plain largely remain unexplored. In the lower valley, three boreholes (Reference number 11233X0124, 11233X0102, 11233X0101; [68] taken

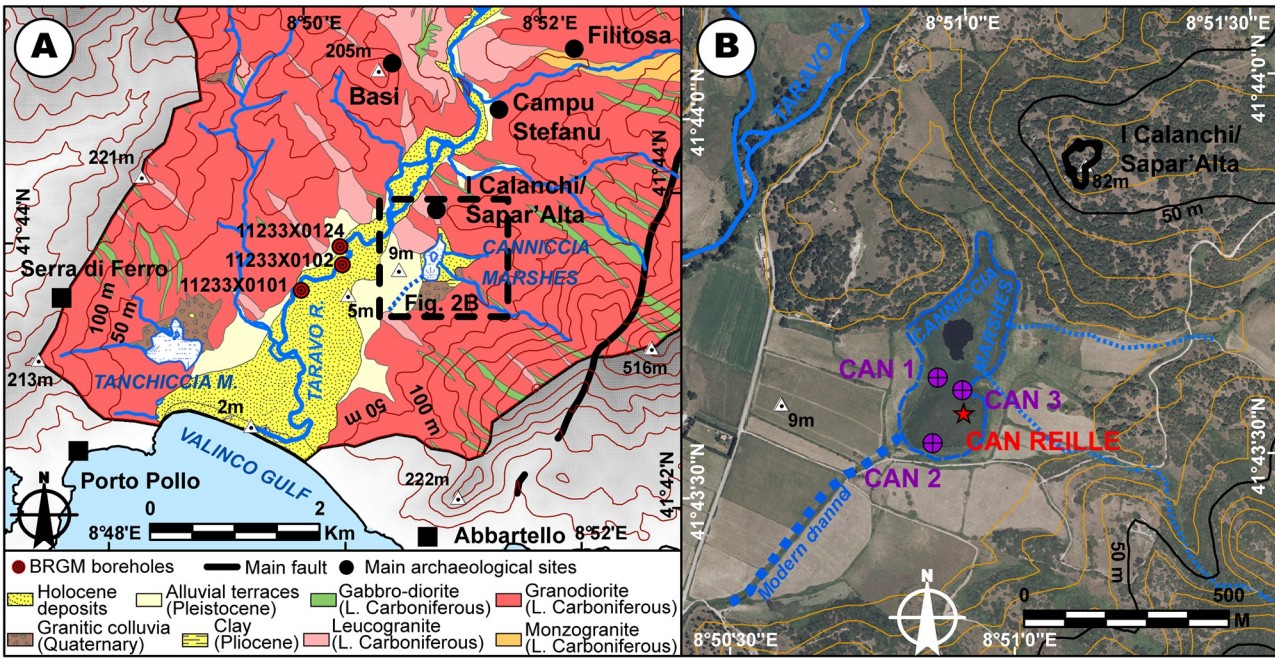

**Fig 2. Sites implantation within the lower Taravo Valley.** The data for the DEM (BdAlti[®]) and orthorectified aerial pictures was acquired from the IGN (Institut National de l'Information Géographique et Forestière; [®]IGN2019) Map Viewer (public domain). Elevation contours (10 m) are derived from [®]BdAlti25 data. The data for the geology was acquired from the BRGM Map Viewer (public domain), modified. **A: Simplified geology of the Lower Taravo Valley and location of Fig. 2B. B: Orthorectified aerial picture of Canniccia marshes**.

in geological surveys revealed only the uppermost stratigraphy (Fig 2A). Despite the lack of dating for the late Holocene Epoch, these results indicate strong fluvial dynamics and highlight different alluviation phases alternating with low-energy deposition phases. On both sides of the valley, two marshes are surrounded by smoothed granodioritic hills and are isolated from Taravo hydrological inputs (Fig 2B).

During the late Holocene (5000 BC), in the Western Mediterranean, in the Rhône [5, 16, 77–78], on the Languedoc coast [17] and around the Gulf of Lion [79] an upsurge in flood, incision and collapse events are clearly demonstrated. These events are associated with an increasing aridity and colder climatic conditions, and major climatic events such as the events 2.2 and 1.2 ka BC among others [80–81]. From a climatic point of view, they are characterized, in the Aegean basin, by a decline in precipitation [21, 82] while the central-western part of Europe seems to be affected by higher humidity [80]. In the Western Mediterranean, the aridity increased torrentiality, flooding and incision [78]. The Medieval Climatic Anomaly (MCA, ~750 to -1000 AD), is characterized, in the Western Mediterranean, by a hydrological deficit and by a low level of high altitude lakes. In the lower valleys, rivers tend to reduce in number of channels and to incise the sedimentary formations of alluvial plains [17, 83]. During the Little Ice Age (LIA), the valleys of the Northern Alps and the Rhone show a significant increase in floods between 1350 and 1850 AD [84–86]. Similarly, on the Roussillon and Lez plains, rivers show a significant increase in sedimentation in the lower valleys [17, 87].

Recent work carried out in Northern Corsica on the filling of wetlands in small watersheds shows an increase in hydrosedimentary processes starting from 5000 BC [43–44]. There is still a limited number of works proposing precise chronologies of detritism and incision during the Holocene phases in Corsica. The combined contribution of the morphosedimentary and

palynological studies of the Canniccia Marshes, in the Taravo Valley allowed to specify the influence of climate and societies on the landscape.

## Present-day climate and vegetation

Mediterranean climate is highly influenced by its relief topography. For Corsica, the mean annual temperatures range between 14.5˚C and 16.5˚C and the mean annual precipitation is 890 mm per year [88–89]. On the coastal area, the mean annual temperature is 17˚C and there is not or, a few, below freezing temperatures during the year [88–89]). From 600 to 1200 m of elevation, the mean annual temperature is between 10˚C and 13˚C. Above 1200 m, winter is marked by abundant precipitation, important daily thermal amplitude and negative temperatures from December through February [88–89]

Corsica is dominated by a siliceous Mediterranean vegetation with evergreen and broad-leaved oak forests respectively dominated by *Quercus ilex* L. and *Quercus pubescens* Willd., maquis with *Erica arborea* L., pine forests with *Pinus pinaster* Aiton and *Pinus. nigra* subsp. *laricio* Maire and euro-siberian forest with *Fagus sylvatica* L. and *Abies alba* Miller [88–89]. The riverine forest comprises mainly alders and *Salix* and, at a low altitude and along intermittent rivers, *Nerium oleander* L. and *Vitex agnus-castus* L. The estuary of the Taravo Valley (where the cores were sampled for the present study) is located in the thermomediterranean stage. The vegetation is characterized by formations of *Pistacia lentiscus* L. along with maquis species (*Arbutus unedo* L., *Erica arborea* L., *Olea europaea* subsp. *oleaster* (Hoffmanns. & Link) Negodi, *Phillyrea angustifolia* L., *Myrtus communis* L. and *Juniperus phoenicea* L.). Between 150 m and 900 m of altitude, typical associations of the mesomediterranean stage are encountered with a sclerophyllous forest composed by *Quercus ilex* L. and *Q. suber* L., maquis species, and, locally broad-leaved oak forests with *Q. pubescens* Willd. and *Castanea sativa* Miller. Dwarf matorrals combined with many species of *Cistus*, Labiatae and Fabaceae are also identified. In the supramediterranean stage (> 900 m of elevation), the pine forest with *Pinus pinaster* Aiton and *P. nigra* subsp. *laricio* Maire and an understorey of *Erica arborea* L. dominate. *Fagus* and *Abies* forests grow in the mountain until an altitude of 1800 m. Above that level, the landscape can be wetter with dense population of *Alnus alnobetula* subsp. *suaveolens* (Req.) Lambinon & Kerguélen, or *matorrals* with *Juniperus*, *Genista*, *Thymus*, *Astragalus* or grasslands with *Armeria*, *Paronychia*, Poaceae, Caryophyllaceae or Scrophulariaceae.

## Sea level variations in Corsica and Northern Sardinia

Post-Glacial Era sea-level rise in the Mediterranean induced repeated shifting in shorelines as well as the position of the estuaries/deltas and river courses [5, 90–91]. Based on the study of algae bioconstructions and shell accumulations, several pioneering works describe sea level variations for the past 30,000 years [92–95]. More recently, synthetic work on the Western Mediterranean has allowed a more precise description of this variation for the late Holocene [96]. The synthesis of the results from major studies on Corsica demonstrate that: 1) Relative Sea-Level (RSL) rose from −33.5 ± 1.6 m at ~8000 BC to −17.5 ± 1.6 m at ~6000 BC during the Mesolithic and Neolithic occupations of the Taravo Valley; 2) Around ~2000 BC, the beginning of Bronze Age, RSL was around −1.5 ± 0.4 m; 3) During Roman Times (~0 BC), RSL was at −0.9 ± 0.5 m; 4) and during Medieval Times (~1000 AD) RSL was located at −0.4 ± 0.5 m. Although Corsica and Sardinia are located in a relatively stable tectonic area of the Western Mediterranean [42], there is a vertical gap between the measurements made in the north (Cap Corse) and further south (Scandola). This divergence would be of tectonic origin [93].

## The mid to late Holocene vegetation history of Corsica

The history of the Corsican Holocene vegetation relies on a set of sites of high, medium and low altitude [41–42, 44, 54, 56–57, 97–98].

**Middle Holocene (6200–2200 BC).** In the Atlantic (Neolithic, 6650–4000 BC), *Erica arborea* maquis is at its optimum at low elevation sites, whereas it is replaced by *Pinus nigra* subsp. *laricio*, deciduous *Quercus*, *Fagus*, *Abies* and *Taxus* at higher altitudes. The Holocene climatic optimum is marked by the expansion of the beech-fir forest at a high altitude, and by the presence of mesophilous forests with deciduous *Quercus* and *Taxus* at medium and low altitudes. In the coastal sites, within the formations dominated by *Erica arborea*, thermophilous Mediterranean woody plants (*Olea*, *Pistacia*, *Arbutus*, *Cistus*) are noted along the abrupt western coasts, while on the lower eastern coasts, on deep soils, the deciduous oak and mixed formations of *Taxus* develop [54].

The Late Middle Holocene (Neolithic-Chalcolithic, 4000–2200 BC) is marked by the optimal expansion of the deciduous *Quercus* forests up to 1300 m altitude. At lower elevations and in the littoral zones, *Quercus ilex* knows a strong expansion to the detriment of *Erica arborea*, deciduous *Quercus* and *Taxus*. These changes would be of human origin as indicated by the discontinuous occurrences of Cerealia [54, 98].

**Late Holocene (2200 BC to present).** The formations with *Erica arborea* regress along the western coasts and expand, on the contrary, on the eastern coast. From 1000 BC, the expansion of *Quercus ilex* continues at a low altitude until it reaches a forest optimum. At the end of the Subboreal, *Quercus ilex* reaches areas up to 1300 m altitude where it is mixed with *Erica arborea*. The beech-fir forest still occupies sites at this altitude. The increase in the percentage of the *Abies* between 1000 and 500 BC would be linked to a decline of fires in Corsica, under colder and wetter climatic conditions [99]. In the Subatlantic Period (1600 BC to the present), major changes in vegetation due to increased human activity occurred as indicated by the increase in cereals, *Artemisia*, *Rumex* or *Plantago lanceolata*. In the mountains, the ultimate degradation of the deciduous oak is recorded between 400 and 600 AD, due to successive fires [99]. From the Genoese Period (1300–1400 AD), a very strong decline of all type of forests (*Quercus ilex*, *Fagus* and particularly *Abies*) occurs everywhere in Corsica to the benefit of land dedicated to agriculture or breeding. The cultivation of *Olea*, *Juglans*, *Castanea* and Vitis was practiced in Corsica during the late Holocene, but the chronological data is unclear.

## Human occupation of the lower Taravo Valley since the Neolithic Age

The lower Taravo Valley has been explored by several research projects since the 1950s. Grosjean [100–101], then Cesari [102–104] revealed numerous archaeological remains testifying to the hierarchy and structure of the pre- and proto-historic occupation of the low valley (Fig 1). Throughout the Neolithic Age, human occupation in the valley was important [100, 104–107], and from the Chalcolithic Age (3600 BC-2200 BC), I Calanchi/Sapar' Alta (Fig 2A and 2B) become one of the most densely populated sites of the Lower Valley [103]. This period is characterized in Corsica by the development of the Terrinian Culture to which is associated at least a dozen of sites spread all over the island [107, 108–112].

The transition from the Late Chalcolithic Age to the Bronze Age (Early Bronze Age, e.g. 2000 BC) is characterized by the fortification of settlements [107], which became widespread in the valley during the Middle Bronze Age (1650–1350 BC) [109]. Numerous examples of such defensive infrastructures are recorded in the valley (Cucuruzzu, Calzola-Castellucciu, Basi, Filitosa, Saparedda or Salvaticu) but I Calanchi/Sapar'Alta (Fig 2A and 2B) contains even more archaic characteristics of the Early Bronze Age (2000–1650 BC) [112]. These fortresses (castelli) are composed of turriform fortified walls of dry stone and houses of irregular plans

**Table 2. Carpological analysis from I Calanchi/Sapar'Alta (from [121] modified).**

| Common name | Species | Macroremains | Chronology | | |
|---|---|---|---|---|---|
| | | | Middle Bronze Age (1800–1300 BC) | Middle to Late Chalcolithic (3000–2100 BC) | Early Chalcolithic (*3090–2921 cal. BC) |
| Barley | *Hordeum vulgare* L. | Seed | + | +++ | + |
| | *Hordeum* sp. | Seed | | ++ | |
| Wheat | *Triticum monococcum* | Seed | | + | |
| | *Triticum dicoccum* | Seed | + | ++ | |
| | *Triticum aestivum/ durum* | Seed | + | ++ | |
| Lentil | *Lens culinaris* Medik | Seed | + | ++ | |
| Bean | *Vicia faba* L. | Seed | + | | + |
| Pea | *Vicia* ssp. | Seed | | ++ | |
| | *Pisum sativum* L. | Seed | | ++ | |
| | *Lathyrus* sp. | Seed | | + | |
| Wild grapevine | *Vitis vinifera* ssp. sylvestris | Grape seed | | ++ | |
| Olive tree | *Olea europaea* L. | Olive core | | + | |

* calibrated dating,

+ presence,

++ frequent,

+++ abundant.

[113]. Agriculture and pastoralism (goat, cattle and pig) were widely adopted and metallurgy was practiced [112, 114–116]. The characteristics of ceramic making and lithics is homogeneous from the Neolithic to the Bronze Ages all over Corsica [101, 107–108]. Most Bronze Age settlements in Corsica are located on a rocky promontory that dominates a floodplain and its ponds/marshes [112, 117]. During the Late Bronze Age (ca. 1350–1200 BC), most of these littoral sites were abandoned while other settlements flourished in interior sectors of the island [109, 113]. Around 1000 BC (Final Bronze Age 1200–900 BC), only main sites (such as Basi, Filitosa) remained occupied [107]. The social transformations attendant on the abandonment of the sites of the lower valley are still uncertain. However, it is during this period that menhir statues were erected. These anthropomorphic warrior stelae, organized to form a network [118–119], are often found along riverbanks, near resurgences of aquifers or temporary marshes [120]. Various macro remains of cultivated plants were discovered at I Calanchi/ Sapar'Alta [121]. They attest to the consumption of barley (*Hordeum vulgare L.*) and beans (*Vicia faba L.*) during the Early Chalcolithic Period (Table 2). The Middle and Late Chalcolithic Periods are characterized by a diversification of food and agriculture. Macro remains discovered include cereals (Barley: *Hordeum vulgare L.*, *Hordeum* sp.; wheat: *Triticum monococcum*, *Triticum dicoccum*, *Triticum aestivum/durum*), leguminous plants (Lentil: *Lens culinaris Med.*; Bean: *Vicia faba L.*; Peas: *Vicia* ssp., *Pisum sativum L.*, *Lathyrus* sp.), wild grape wine (*Vitis vinifera* ssp. *sylvestris*) and wild olive trees (*Olea europaea L.*). Finally, the Middle Bronze Age showed a decrease in the diversity of cultivated species.

During the first (900–500 BC) and second Iron Ages (Etruscan period 500–250 BC), few settlements remained occupied and some rock shelters were used for burial in I Calanchi/ Sapar'Alta [103] (Fig 2A and 2B). The Roman Period (250 BC-450 AD) does not present any notable occupation of this sector. The end of the Roman influence on the island is followed by centuries of invasion which have no consequences on the agropastoral activities of the lower

Taravo Valley. From 1100 BC, Corsica was at the center of a major territorial challenge between three major powers of the Western Mediterranean: Pisa, Genoa and Aragon [122]. In the lower valley of Taravo, the medieval settlement is represented only by the current village of Serra di Ferro.

This chronocultural context raises several questions. First of all, what were the environmental conditions and agropastoral activities between the Neolithic and Middle Bronze Ages? Similarly, reasons for the abandonment of sites in the Lower Taravo Valley during the Late Bronze Age remain to be explained. This article proposes to put in perspective these societal transformations in relation to the sedimentary and pollen filling of Canniccia Marshes. The results allow us to link landscape evolution to agropastoral activities and climate change.

## Material and methods

Canniccia Marshes and I Calanchi/Sapar'Alta archaeological site are a part of a protected ecological area (Plage et zone humide du bas Taravo et de Tenutella, ZNIEFF: 940004127; Embouchure du Taravo, plage de Tenutella, étang de Tanchiccia, Natura 2000: FR9400610). The field site access was approved by the local delegation of Ministry of Culture (DRAC) and the Environment Office of Corsica.

In order to restore the evolution of the landscape and cultures during mid-Holocene we adopted an innovative interdisciplinary approach associating the methods of stratigraphy and palynology (S1 Text). The sedimentary profiles cored in 2008 were sampled at ~5 cm resolution, yielding 325 samples in total. Grain-size analyses, loss on ignition (LOI) and magnetic susceptibility measurements helped to precise the stratigraphy of the Canniccia Marshes. Only the first 10 m of the core CAN REILLE (S1 Fig) are presented in this article in order to compare the cultural event recorded in this core with the sedimentological events recorded in the rest of the cores of Canniccia. Eighteen samples of peat, plant remains, charcoal and bulk organic sediment were collected in order to obtain a robust chronostratigraphic sequence (Table 3).

## Results

### Chronology

The chronology of sedimentary deposits and palynological records is based on 18 radiocarbon datings presented in Table 3. The dating methods are described in the supplementary information (S 2). The age-depth model of the pollen profile CAN REILLE (Fig 3) was reconstructed using the age modelling method of Blaauw (CLAM R R 2.13.2 software).

[119]. The reconstructed ages between the earliest radiocarbon dating and the surface for each sequences (CAN REILLE, 1262–978 BC; CAN 1, 214–385 AD; CAN 2, 795–542 BC; CAN 3, 2459–2206 BC) are hypothetical as the surface was not dated and is supposed to be actual.

### General stratigraphy

On 2008 cores (Figs 4, 5 and 6), grain size analysis, the concentration of magnetic particles and the OM rates (see methods in the supplementary information S1 Text) allow us to distinguish 10 different Morpho Sedimentary Units (MSU). No sedimentological measurements were carried out on CAN REILLE. Stratigraphy and main sedimentary features of the MSU are presented from bottom to top in Table 4. They show an alternation of phases of destabilization marked by detrital episodes (D1 to D4) and phases of stabilization marked by marshy environments (T1 to T4). Characteristics of detrital events suggest colluvial and local sedimentary

**Table 3. Radiocarbon dating results.**

| Sample name | Depth (in m) | Elevation above mean sea-level (in m) | Material | Laboratory reference | Age in BP | Error ± | Age cal. AD 2 σ | Age cal. BP |
|---|---|---|---|---|---|---|---|---|
| CAN1C3 205cm | 2,05 | +6,95 | Peat | Saca-30518 | 1755 | 30 | 214–385 AD | 1650 |
| CAN1C3 260cm | 2,6 | +6,4 | Peat | Saca-30519 | 2220 | 30 | 374–203 BC | 2240 |
| CAN1C4 205cm | 3,05 | +5,95 | Peat | Saca-30520 | 2595 | 30 | 823–758 BC | 2740 |
| CAN2C2 195cm | 1,95 | +7,05 | Peat | Poz-49384 | 2525 | 30 | 795–542 BC | 2620 |
| CAN2C3 258cm | 2,58 | +6,42 | Peat | Poz-49383 | 2785 | 30 | 1008–844 BC | 2875 |
| CAN2C5 462cm | 4,62 | +4,38 | Plant remains | Poz-49381 | 2895 | 35 | 1208–992 BC | 3050 |
| CAN2C6 549cm | 5,49 | +3,51 | Plant remains | Poz-49380 | 3025 | 30 | 1394–1193 BC | 3245 |
| CAN2C6 583cm | 5,83 | +3,17 | Peat | Poz-49386 | 2995 | 30 | 1383–1116 BC | 3200 |
| CAN2C6 590cm | 5,9 | +3,1 | Peat | Poz-49377 | 3040 | 35 | 1409–1208 BC | 3260 |
| CAN2C7 638cm | 6,38 | +2,62 | Peat | Poz-49376 | 3130 | 35 | 1462–1296 BC | 3330 |
| CAN2C7 658cm | 6,58 | +2,42 | Peat | Poz-49374 | 3125 | 35 | 1494–1289 BC | 3340 |
| CAN2C7 680cm | 6,8 | +2,2 | Charcoal | Poz-49405 | 3225 | 30 | 1606–1429 BC | 3470 |
| CAN3C3 235cm | 2,35 | +6,65 | Peat | Saca-30521 | 3850 | 30 | 2459–2206 BC | 4280 |
| CAN REILLE TII 95 | 0,95 | +8,05 | Bulk organic sediment | AA#25296 | 2920 | 50 | 1262–978 BC | 3066 |
| CAN REILLE TII 320 | 3,2 | +5,80 | Bulk organic sediment | AA#25297 | 3500 | 50 | 1944–1729 BC | 3771 |
| CAN REILLE TII 610 | 6,1 | +2,90 | Bulk organic sediment | AA#25298 | 3675 | 55 | 2202–1916 BC | 4010 |
| CAN REILLE TII 920 | 9,2 | -0,2 | Bulk organic sediment | AA#25299 | 3850 | 70 | 2487–2133 BC | 4269 |
| CAN REILLE TII 1000 | 10 | -1 | Bulk organic sediment | AA#25300 | 4385 | 55 | 3119–2897 BC | 4990 |

inputs during a reactivation of an alluvial cone, as well as fluvio-torrential dynamics with overflow conditions.

## Pollen data

The pollen data published in this article correspond to the upper part of an unpublished diagram [65] (S1 Fig). In this upper part, seven Local Pollen Assemblage Zones (LAPZ) have been distinguished on the basis on the fluctuation of Arboreal Pollen (AP) and indicators of agricultural activities (Fig 7). The main characteristics of the LPAZ are shown in Table 5. The pollen diagram shows a very fast alternation of AP and NAP (non-arboreal taxa). These rapid fluctuations mainly concern maquis trees, riparian forest taxa and herbaceous taxa related to human activity.

## Discussion

### Morphosedimentary evolution of Canniccia Marshes from mid Chalcolithic (3000 BC) to modern times

Stratigraphical studies, combined with grain size analysis, Magnetic Signal (MS) and OM content, allow us to reconstruct the morpho-sedimentary dynamics of Canniccia Marshes since 3000 BC (Mid Chalcolithic Period). The three cores highlight four phases of intense detritism. Magnetic signal variations of these phases raises the question of the origin of the fluvio-torrential and colluvial deposits. Finally, four phases of sedimentary stabilization are identified attesting to the development of marshy environments (Fig 8).

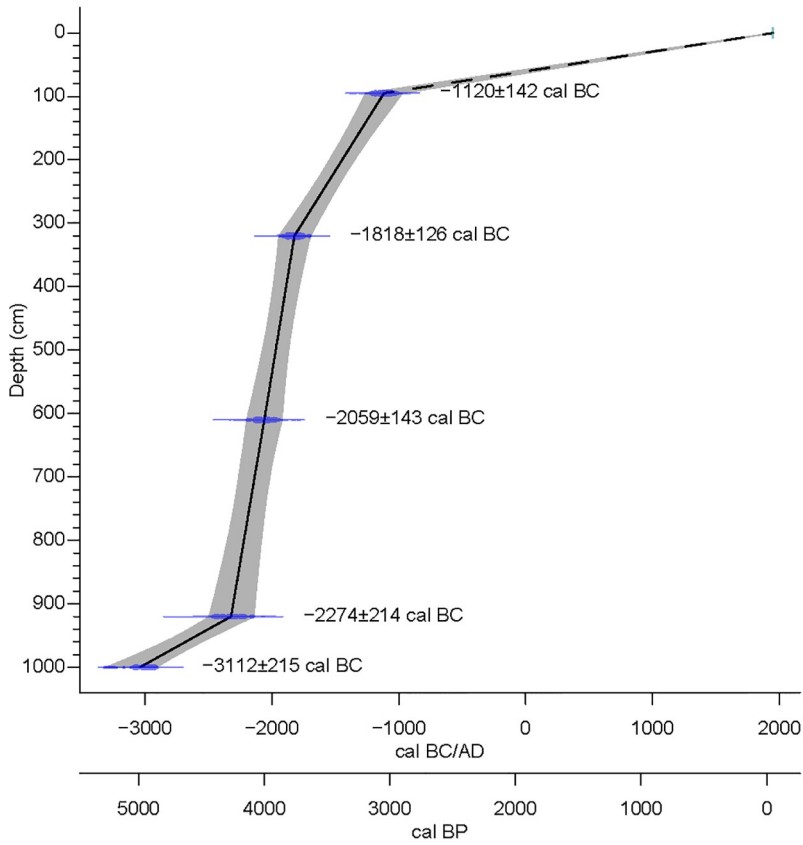

**Fig 3. Age-depth model of the upper 10 m of the palynological profil CAN REILLE.** The age-modelling is based on the method of Blaauw (2010) (CLAM R R 2.13.2 software) [123].

**Four detritic phases identified on Canniccia Marshes since 3000 BC.** Morphosedimentary units have a fan-shaped organization thickening as one moves away from the eastern slope of the Taravo Valley (Fig 8). Four detritic events have been identified.

D1 appears in CAN 3 only. It is composed of angular granite pebbles and sands. The end of this fluvio-torrential event is dated to the Mid Chalcolithic Age (~ 3000 BC). D2 consists of a main channel near CAN 2 (angular granitic pebbles) and overflow deposits in CAN 3 (silt and fine sand). The small thickness of the sediments in CAN 3 does not allow for a high resolution of these events. It testifies, however, to relatively high accumulation rates (45 cm per century) at the transition between the Chalcolithic and Early Bronze Ages. This torrential event is dated between 2300 and 1700 BC. D3 is the best detrital event documented in the Canniccia Marshes. It testifies to the establishment of fluvio-torrential conditions with a main channel near CAN 1. The morphology of this channel can be followed in all the cores. D4 is composed of angular granitic pebbles in CAN 1, an alternation of fine and very fine sands in CAN 2 and clayey silts in CAN 3. This event is dated to the Final Bronze Age (between 1200 and 1000 BC) and it attests to particularly high sedimentation rates (75 cm per century). On the opposite bank of the Taravo (Figs 1C and 2A), the Tanchiccia Marshes present a chronostratigraphical succession which testifies to a history of storms in this coastal context (1700–1500 BC) and local colluvial deposits (1600–1400 BC) [42]. A detrital event identified between 3500 and 2700 BC in these marshes can be associated with the D1 event in the Canniccia Marshes. In the Western Mediterranean, paleoclimatic reconstructions demonstrate the existence of

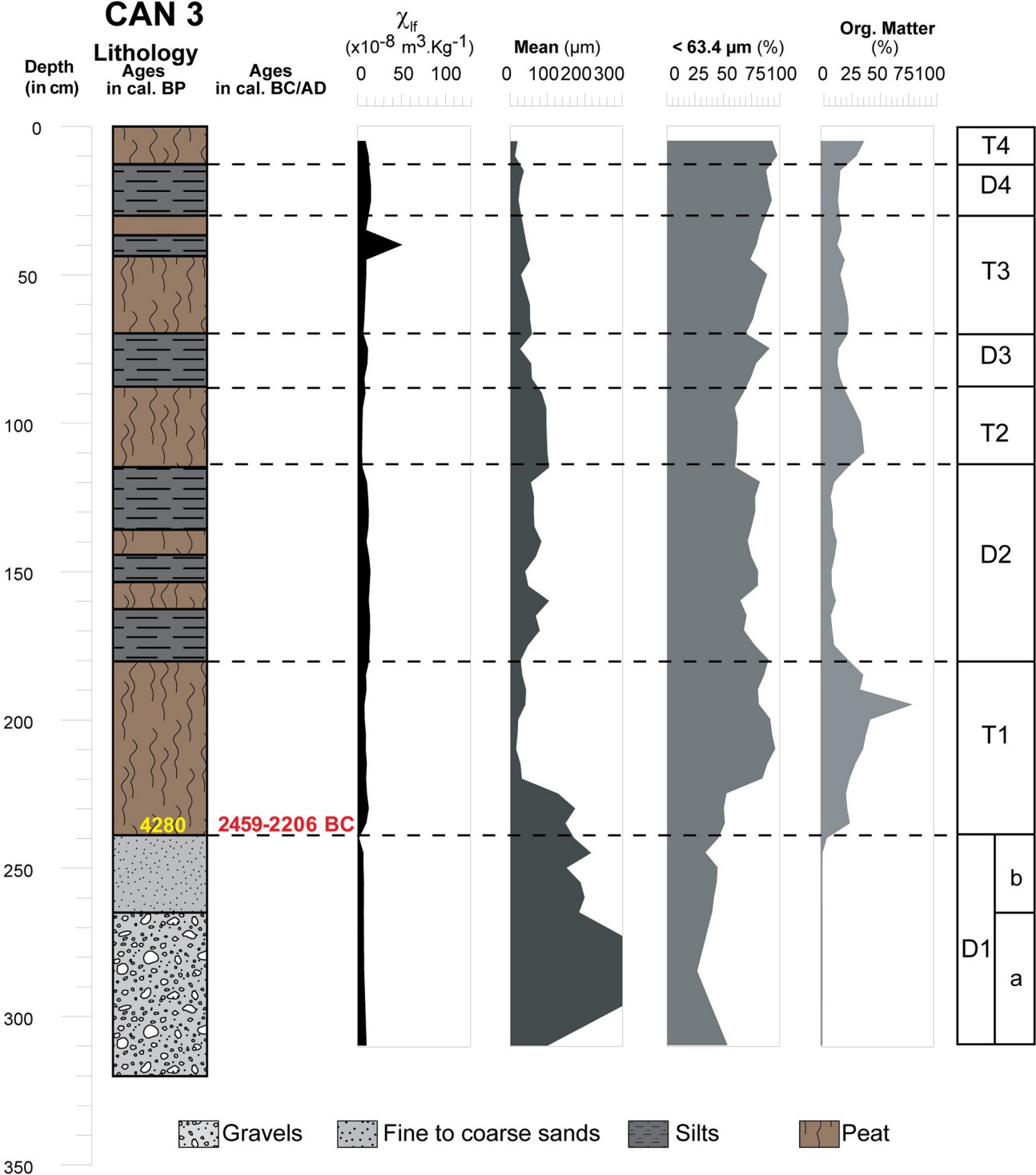

**Fig 4. Core profile of CAN 3 borehole where sedimentological results and environments are presented.** Magnetic susceptibility measurement at low frequency, LASER grain size and organic matter content.

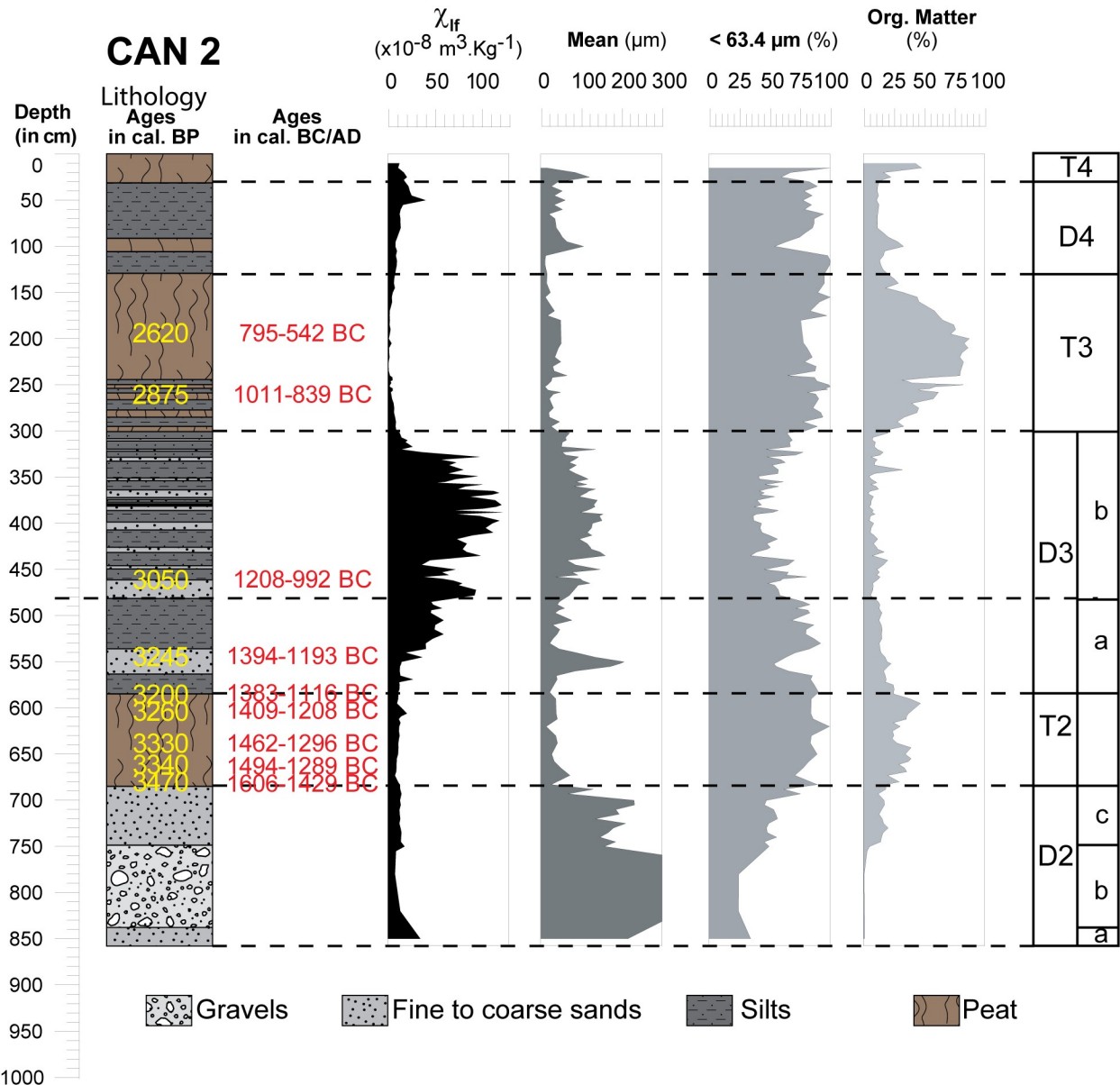

**Fig 5. Core profile of CAN 2 borehole where sedimentological results and environments are presented.** Magnetic susceptibility measurement at low frequency, LASER grain size and organic matter content.

cooling and increased aridity during the Holocene Epoch [80–81, 124–126]. The detrital event D1 (around 3000 BC) in the Canniccia marshes does not correspond to any major climatic events recorded in the Western Mediterranean. The possibility of a relationship between D2 and D3 in regard to increased detritism in the Canniccia marshes with 2.2 and 1.2 ka BC events is therefore plausible. An event that was coeval with D2 is also observed in the wet plateau of Cauria in the south of the island [120]. This event could have affected the entire island.

We also notice that sedimentation rates are low for medieval times and do not bring to light the Medieval Climatic Anomaly (MCA ~750–1000 AD, [17, 83], whether the LIA (~1350–1850 AD) identified in Northern Corsica [44] or in the Rhone [84–86, 127].

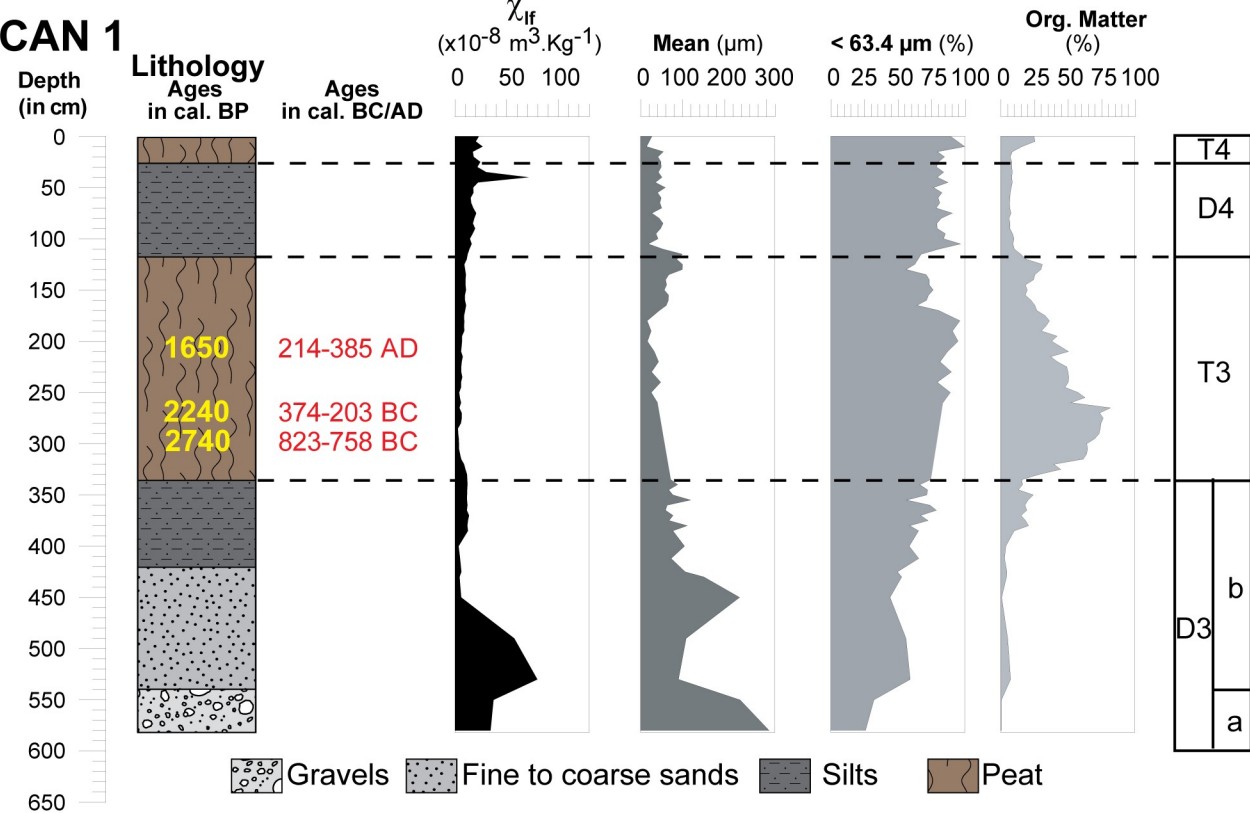

**Fig 6. Core profile of CAN 1 borehole where sedimentological results and environments are presented.** Magnetic susceptibility measurement at low frequency, LASER grain size and organic matter content.

**Origin of deposits.** The detrital events of D1 had a low magnetic susceptibility ($\sim$10 $\times 10^{-8}$m$^3$.kg$^{-1}$) (Fig 9). D2, D3 and D4 have different characteristics. In D2, we note a weak increase of the magnetic signal between the coarse formations ($\sim$10 $\times 10^{-8}$m$^3$.kg$^{-1}$) at the bottom, and the sands ($\sim$20 $\times 10^{-8}$m$^3$.kg$^{-1}$) at the top. This event is not observed in CAN 3. Similarly, D3 presents comparable characteristics to those suggested by D2. In CAN 1, pebbles and gravels (D3a) show MS measurements around 35 $\times 10^{-8}$m$^3$.kg$^{-1}$. For D3b, we notice a significant increase in the magnetic signal for fine sand (MS up to 50 $\times 10^{-8}$m$^3$.kg$^{-1}$). The dilation of deposits in CAN 2 allows us to provide more details on this enrichment in magnetic grains during D3. D3a is composed of silty to fine sand with a medium magnetic signal (10< MS <50 $\times 10^{-8}$m$^3$.kg$^{-1}$). D3b is characterized by alternating centimetric beds of fine and very fine sand which shows a significant increase in MS measurements (40< MS <120 $\times 10^{-8}$m$^3$.kg$^{-1}$). Highest measurements are obtained for mean grain size between 110 and 160 μm. Finally, in CAN 1 and CAN 2, we notice a weak enrichment in magnetic grains at the end of the D4 event. Magnetic signal decrease is associated with high OM content, while its increase is clearly related to the detrital origin of the deposits. In general, granodiorites and monzogranites (75–90 $\times 10^{-8}$m$^3$.kg$^{-1}$) as well as Pliocene Terraces (10 $\times 10^{-8}$m$^3$.kg$^{-1}$) that compose the catchment basin of the Canniccia Marshes exhibit medium to weak magnetic signals [42, 44–46]. On the other hand, in the middle Taravo Valley, diorite, gabbro and greenstone rock veins reveal the highest values of MS (between 250 and 4000 $\times 10^{-8}$m$^3$.kg$^{-1}$). D1 and D2 have low magnetic susceptibility values and thus attest to mainly local and colluvial inputs (MS<20 $\times 10^{-8}$m$^3$.kg$^{-1}$) and seem to exclude a fluviatile origin. D3 and D4 show a significant increase in the

**Table 4. Main morphosedimentary units (MSU) characterizing the stratigraphy of CAN 1, CAN 2 and CAN 3.**

| MSU | Sub MSU | CAN 1 (Fig 6) | | | CAN 2 (Fig 5) | | | CAN 3 (Fig 4) | | | Main sedimentary features |
|---|---|---|---|---|---|---|---|---|---|---|---|
| | | Mean | MS | OM | Mean | MS | OM | Mean | MS | OM | |
| | | μm | .10$^8$m$^3$.kg$^{-1}$ | % | μm | .10$^8$m$^3$.kg$^{-1}$ | % | μm | .10$^8$m$^3$.kg$^{-1}$ | % | |
| D1 | D1a | - | - | - | - | - | - | >300 | ~10 | <1 | Coarse elements / Low MS |
| | D1b | - | - | - | - | - | - | ~180 | ~10 | <1 | Low OM |
| T1 | | - | - | - | - | - | - | 20–40 | ~10 | ~40 | Clay / Low MS / High OM |
| D2 | D2a | - | - | - | ~250 | 40 | <1 | 30–60 | 15 | ~20 | Sand / Low SM / Low OM |
| | D2b | - | - | - | >2000 | <10 | <1 | | | | Coarse elements / Low MS / Low OM |
| | D2c | - | - | - | ~80–250 | ~ 25 | ~20 | | | | Fine sands / Medium MS / Medium OM |
| T2 | | - | - | - | ~63.3 | ~10 | 25–35 | 40–60 | <10 | ~ 25 | Silty clay / Low MS / High OM |
| D3 | D3a | >300 | 35 | <1 | 40–200 | 10–50 | ~15 | ~60 | ~50 | <1 | Coarse elements / Medium MS / Low OM |
| | D3b | 100–220 | 80 | <1 | 50–140 | 40–120 | <1 | | | | Fine sands / variable MS / Low OM |
| T3 | | 20–60 | <10 | 80 | 20–60 | <10 | 80 | 20–60 | <10 | 80 | Clay / Low MS / High OM |
| D4 | | 20–100 | 10–70 | 10–25 | 20–100 | 10–70 | 10–25 | 20–100 | 10–70 | 10–25 | Clay / High MS / Low OM |
| T4 | | 20–40 | 15 | 25–40 | 20–40 | 15 | 25–40 | 20–40 | 15 | 25–40 | Clay / Low MS / High OM |

Abbreviation: MS: Magnetic Susceptibility, OM: Organic Matter content.

concentration of magnetic grains suggesting a sediment input from the Taravo. More specifically, the overall results of MS measurements show that the values obtained in D3 are generally higher than those delivered by the rocks of the catchment area from the Canniccia Marshes. It is therefore likely that the middle part of the watershed is the geographical origin of the magnetic mineral recharge of the Taravo floods. Further studies are yet to be carried out on sedimentary archives of the median sector of the Taravo River in order to specify the origins of these erosions periods.

**Four phases of morphosedimentary stability identified in the Canniccia Marshes since 3000 BC.** Between these periods of fluvio-torrential events, four phases of peat formation are

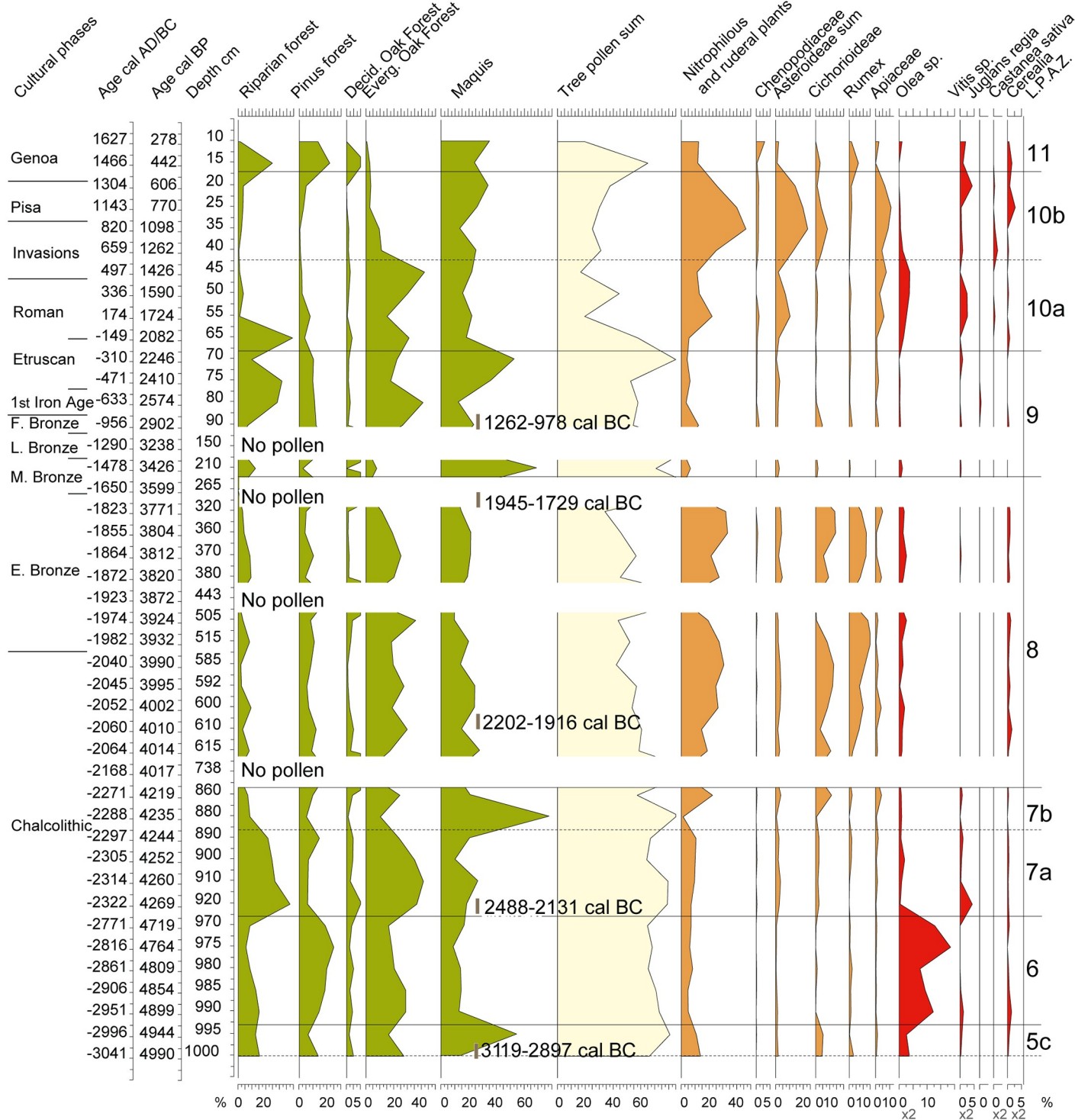

**Fig 7. Simplified pollen diagram of CAN REILLE related to lithostratigraphy.** Time framework covers Subboreal and Subatlantic, i.e. the last 5000 years. Values are in frequencies calculated on a pollen sum without *Alnus*, *Salix*, Cyperaceae, *Sparganium-Typha*, *Ranunculus*, all aquatic plants (*Myriophyllum*, *Nymphaea*, *Potamogeton*, *Utricularia*) and ferns (*Isoetes*, Monolete spores, *Ophioglossum*, *Osmunda*, *Polypodium*, *Pteridium*, *Selaginella*, Trilete spores). Riparian forest (*Alnus*, *Fraxinus*, *Salix*), Pinus forest (*Pinus nigra* subsp. *laricio*, mediterranean *Pinus*), Deciduous oak forest (*Acer*, *Buxus*, deciduous *Quercus*, *Hedera*, *Ilex*, *Prunus*, *Rubus*, *Tilia*, *Ulmus*), Evergreen oak forest (*Quercus ilex*, *Quercus suber*), Maquis (*Arbutus unedo*, *Cistus*, *Erica arborea*, *Erica terminalis*, *Juniperus*, *Phillyrea*, *Pistacia*), Nitrophilous and ruderal plants (*Artemisia*, Apiaceae, Asteroideae, *Carlina*, *Centaurea*, Chenopodiaceae, Cichorioideae, *Plantago coronopus*, *Plantago lanceolata*, *Plantago major/minor*, *Rumex*), Asteroïdeae sum (Asteroideae, *Centaurea*).

**Table 5. Main palynological features characterizing the LPAZ of CAN REILLE.**

| LPAZ | Age cal. BC/AD | Main palynological features |
|---|---|---|
| 5 | 3041–2996 | Maximum of plants of the maquis, mainly *Erica arborea* and maximum of AP |
| 6 | 2996–2771 | Collapse of the maquis taxa, increase of *Pinus*, maximum of *Olea* (18,6% of the PS) |
| 7 | 2771–2271 | Collapse of *Olea*. 7a: maximum of the evergreen oak forest (41,4% of the PS), mainly *Quercus ilex*; maximum of the riparian forest (37,4% of the PS), mainly *Alnus*. Increase of AP. 7b: transitory peak of the maquis (77,9% of the PS); decline of the riparian forest |
| 8 | 2271–1650 | Decline of AP and increase of pastoralism indicators: nitrophilous plants (33,3% of the PS), Cichorioideae (14,3% of the PS), *Rumex* (15,8% of the PS) |
| 9 | 1650–310 | Increase of AP to a maximum (85% of the PS). Collapse of pastoralism indicators. |
| 10 | 310–1304 | Collapse of AP. 10a: Increase of pastoralism indicators (nitrophilous plants, Asteroideae, Apiaceae) and transitory peak of *Olea* (3,8% of the PS). 10b: collapse of the evergreen oak forest (2,7% of the PS); maximum of indicators of pastoralism, mainly nitrophilous plants (46,6% of the PS) and Asteroideae (15,7% of the PS). Short peaks of *Castanea*, Cerealia and *Vitis*. |
| 11 | 1304–1627 | Last transitory maximum of AP (mainly maquis, *Pinus* and riparian forests). Decline of agriculture markers. |

Abbreviation AP: Arboreal Pollen; LPAZ: Local Pollen Assemblage Zone; PS: Pollen Sum.

related to periods of sedimentary stability (T1 to T4). Palynological data of CAN REILLE allow us to specify the chronology of morphosedimentary transformations, in particular for D4 and T4, where we do not have radiochronological dating.

T1 (3000–2300 BC) attests to the predominance of marshy environments during 700 years at the end of Late Chalcolithic Age. At the end of the Middle Bronze Age, T2 (1700–1200 BC), evidence demonstrates that for 500 years, marsh environments prevailed. Episodes of severe storms recorded between 1700 and 1500 BC in Tanchiccia [42] are not identified in Canniccia. This attests to a relative disconnection of Canniccia from coastal events. T3 (1000 BC- 500 AD) can be identified between the Late Bronze Age and Roman Times. This period of stability is the longest recorded in the Canniccia Marshes (~1500 years). Finally, T4 concerns recent periods, probably related to the LIA (1150–1850) identified in Northern Corsica [43–44] and Southern France [127].

## Human occupation and agropastoral activities in the lower Taravo Valley

Pollen records described in the present work cover periods in which human activities have a direct impact on the vegetation throughout Corsica [54, 56–57]. The Creno sequence study (Creno, Fig 1B) demonstrated the role of fire in the past vegetation changes and confirm an increase of fire frequency since 2500 BP [58]. We observe a similar evolution in the lower Taravo Valley Our work confirm to the intensity of Chalcolithic and Early Bronze Age agropastoral activities on the vegetation probably driven by the slash-and-burn agriculture. Moreover, our work allows us to specify the chronological context of these events and to identify the agropastoral activities at the origin of the anthropic pressure on vegetal landscape (Figs 7 and 10).

**The chacolithic age.** Around 3000 BC (LPAZ 5), the vegetal landscape was occupied by a forest dominated by *Erica arborea* (maquis) and *Quercus ilex* (Fig 7). Biological signs (*Olea*, nitrophilous plants) indicating the presence of farmers are present but not significant. Between 3000 and 2500 BC (LPAZ 6), clearing practiced by the human populations living in the lower Taravo Valley, reduced drastically the maquis whose palynological values collapse from 55% to 15% of the PS, to the benefit of olive cultivation whose frequencies reach 14% of the PS at the end of this agricultural event.

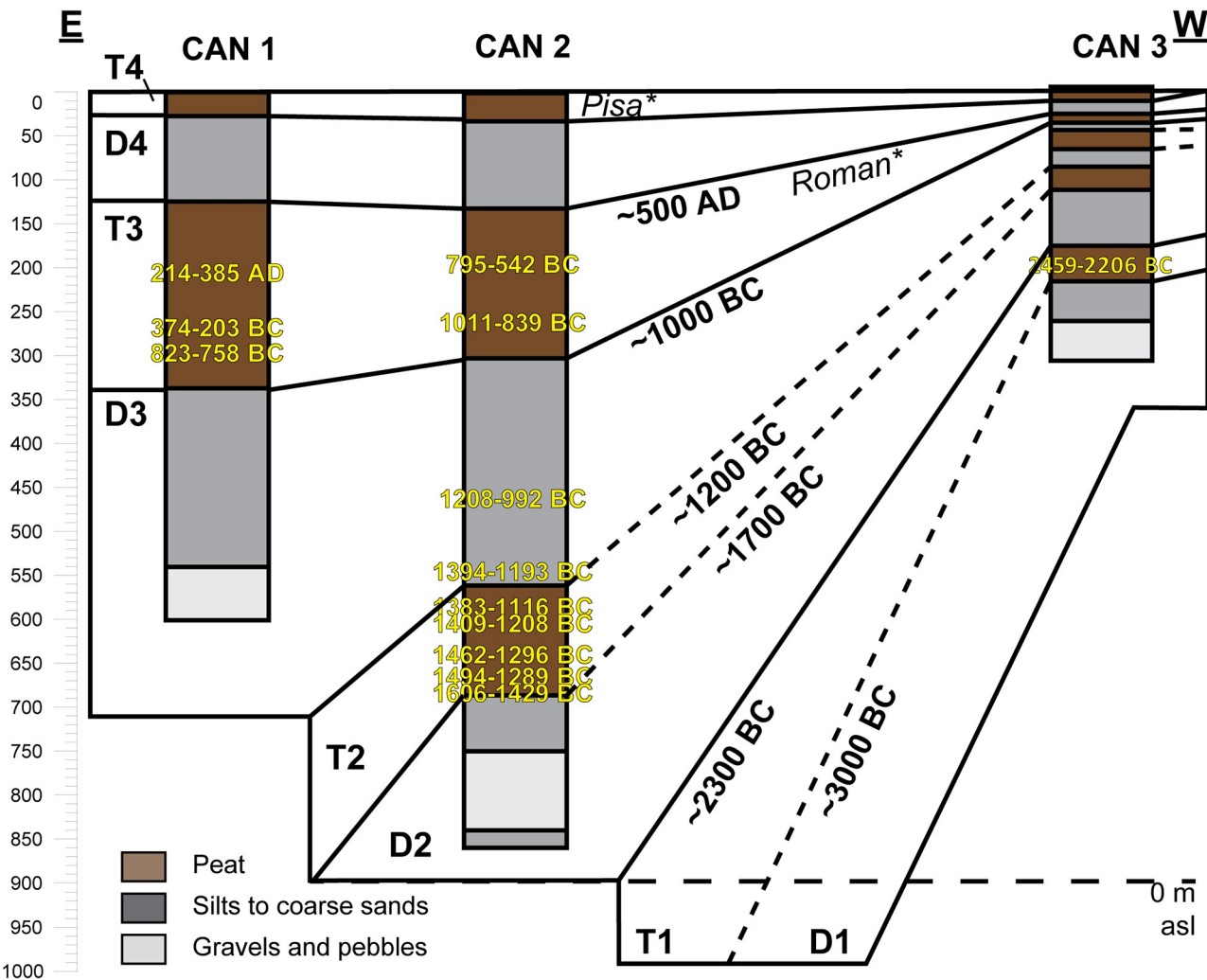

**Fig 8. Stratigraphical correlations between CAN 1, CAN 2 and CAN 3.** Cultural periods, indicated with an asterisk, are derived from palynological analysis.

In Corsica, the pollen of *Olea* sp. is recorded in high mountain lakes (Fig 1 Creno and Bastani) at long distances and attests to the presence on the island of this tree since ~5600 BC. The lower valley ponds (Fig 1 Fango, Sale and Saleccia) also reveal three events where *Olea* sp. was particularly important between 3000–450 BC, around 100 AD and around 1500 AD [54]. Zohary and Spiegel-Roy [128] proposed that olive cultivation appeared around 3500 BC in Palestine. It would have gradually spread to Greece around 2500 BC and then to other parts of the Mediterranean by the Phoenicians, Etruscans, Greeks and Romans [129]. In Northwestern Mediterranean areas, introduction of olive cultivation is attested to by the discovery of the most ancient oil mill dated to the Greeks and then to Romans [130–132]. More recent studies demonstrated that during the Neolithic in the western Mediterranean, *Olea* sp. became dominant in the coastal plant formations [133]. They also revealed several domestication centres [134] and underlined that during the Late Bronze Age, the role of wild olive populations in the plant economy was important [135]. Our study indicates that olive was cultivated in Corsica during the Chalcolithic Age, and that cultivation peaked in this period. The discovery of olive

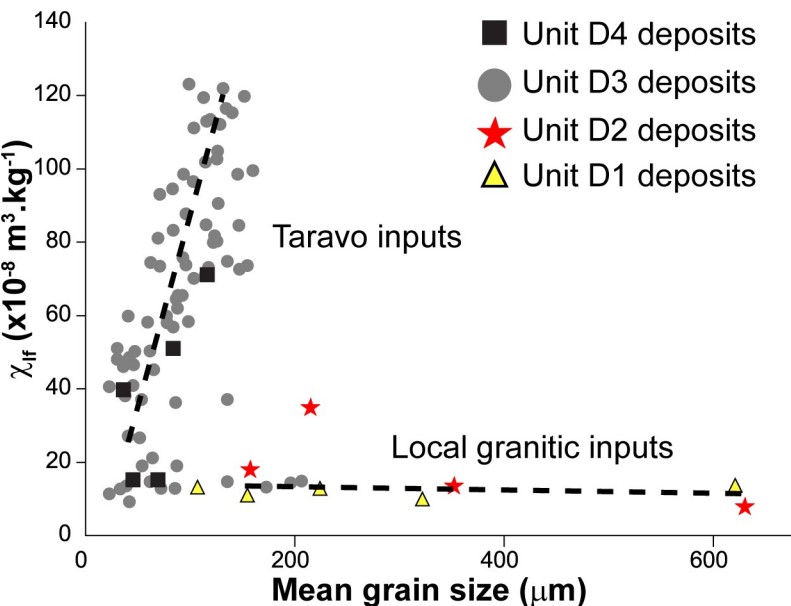

**Fig 9. Relationship between magnetic susceptibility and grain-size distribution from detrital sediments sampled on boreholes CAN 1, CAN 2 and CAN 3.**

stones on the archaeological site of Scaffa Piana in Northern Corsica dated to 2825 BC [136] seems to confirm this hypothesis.

Plant macroremains found in the prehistoric sites of I Calanchi/Sapar'Alta [121] (Table 2) dated to the Chalcolithic Age show that many species of cereals (*Hordeum vulgare*, *Hordeum* sp., *Triticum monococcum*, *T. dicoccum*, *T. aestivum/durum*), Fabaceae (*Lens culinaris*, *Vicia faba*, *Vicia* ssp., *Pisum sativum*, *Lathyrus* sp.), and vine (*Vitis vinifera*) were also cultivated in the lower Taravo Valley that was, at that time, densely populated. A diversity of agricultural and pastoral activities testify to the diversity of the diet of the human populations present in the lower Taravo Valley at this moment.

Cereals were introduced in Corsica during the Neolithic Age. Their cultivation is evidenced by the presence of millstones and seeds found at archaeological sites [112, 115, 117] and by pollen data from littoral sites on the east side of the island [56]. In the Canniccia Marshes, cereals were present already in the base of the diagram and thus attest to their cultivation since at least 3000 BC in the lower Taravo Valley.

The late Chalcolithic, between 2500 and 2200 BC (LPAZ 7), corresponded to an abandonment phase characterized by a collapse of olive cultivation and by the reinstallation of the evergreen oak forest on the slopes and of the riparian forest along the Taravo. Vines seem to be the only plant to be transitorily cultivated at the beginning of this period (2300 BC). This abandonment phase is contemporary to the 2.2 ka BC event, a complex climatic event considered to feature colder and either drier or wetter winters than during the previous period, depending on the region [137].

Viticulture would have started between ~5000–4000 BC in Greece and Crete [138]. Phoenician influence (1000–500 BC) appears to have played a significant role in the establishment and diffusion of viticulture and viniculture in the Western Mediterranean [139–140]. In Italy, most ancient testimonies of vine cultivation date back to 900–800 BC [141]. Finally, the emergence of viticulture in France was concomitant with the foundation of Marseille (~600 BC) by the Phocaeans [142]. Studies on seeds and pollen are still too limited to differentiate between

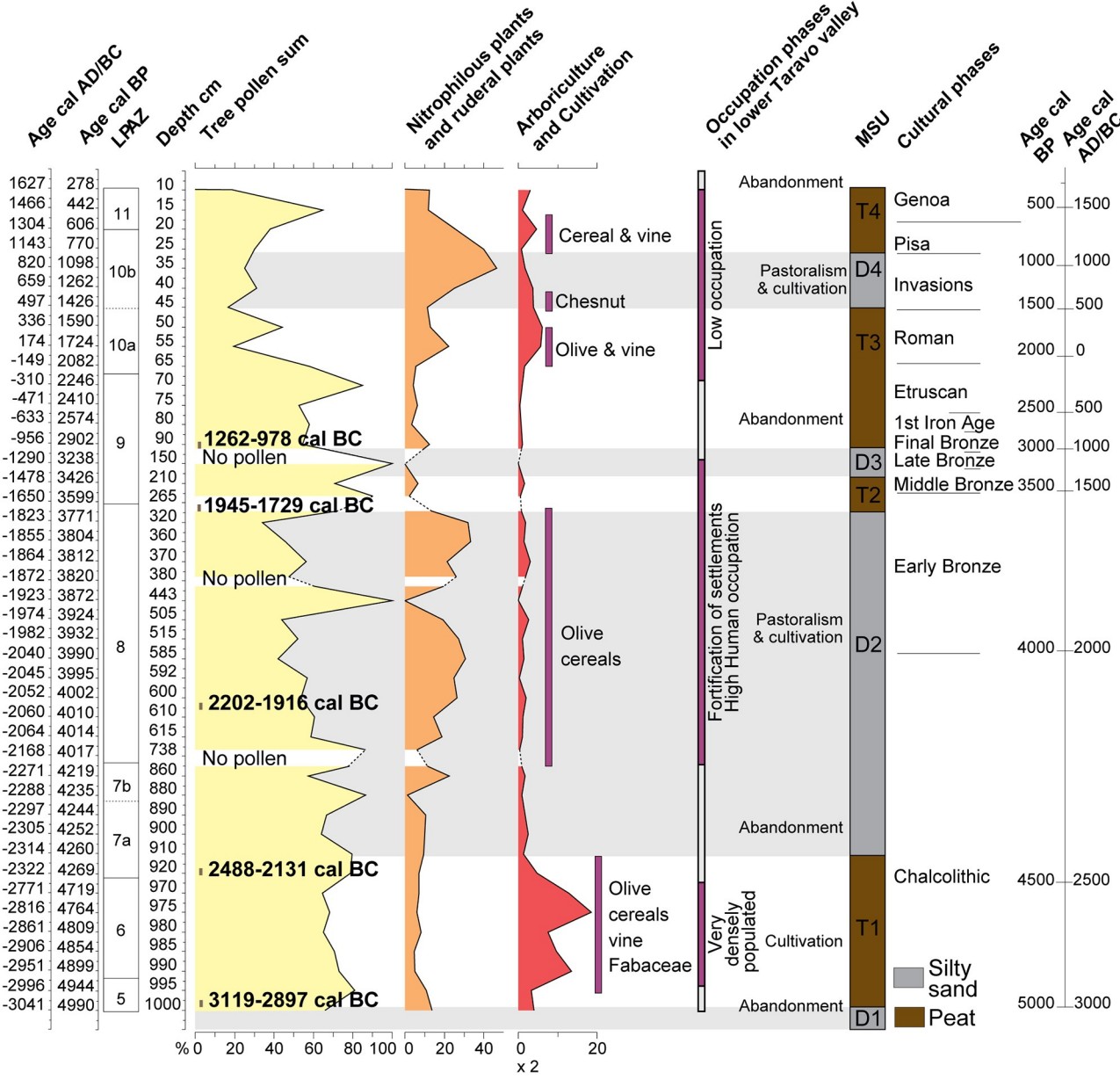

**Fig 10. Synthetic anthropogenic diagram inferred from palynological records cross combined with general stratigraphy.**

the wild and cultivated *Vitis* sp. [141, 143–147]. More recent studies attest to a primitive culti-vated *Vitis vinifera* in Sardinia during the Late Bronze Age [148]. In Corsica, the study of pol-len filling high mountain and low valley lakes testifies to the punctual presence of *Vitis* sp. since at least 3000 BC (Fig 1, Bastani, del Sale) [56]. In the Canniccia Marshes, the pollen of *Vitis* sp. has been identified in association with *Olea* sp. between 3000 and 2300 BC, during a period when the lower Taravo Valley was intensely populated (Figs 7 and 10). The origin of the domestication of olive tree and vineyard remains therefore uncertain in the lower Taravo Valley. However, because of the low pollen production of *Vitis* and its low ability to disperse its pollen, it seems possible that the pollen of *Vitis* is of local origin and comes from either wild or cultivated plants.

**From the Bronze to the Genoa Periods.** During the Early and the Middle Bronze Ages (from ca 2000 to ca 1700 BC LPAZ 8), a major transformation of agrarian activities is recorded (Figs 7 and 10). A strong increase of nitrophilous plants is indicated, as well as the cultivation of olives and cereals. The forest declined, as indicated by the decrease of arboreal pollen. It is interesting to note that the forest which survived was the riverine one. As indices of pastoralism are high, we suggest that the decline of the riparian forest may be related to intensive pastoralism. This period is characterized by the fortification of the settlements.

This phase is interrupted by a new period of abandonment (from ca 1500 to 200 BC, LPAZ 9) that is mainly marked by a decline of cultivation and pastoralism, but the continuity of the pollen curves of the nitrophilous and cultivated plants show that the agricultural activities did not completely stop. The decline in anthropogenic pressure allowed the forest to re-establish itself, in particular the riparian forest, but also the maquis whose frequencies reach a maximum (55% of the PS) at the end of this event. Archaeological data indicate that this event also corresponded to a phase of abandonment. Chronologically, this period goes from the late Bronze to the Etruscan Phases.

As mentioned above, the age model of the last 2000 years is weak as the surface of the core was not dated. This is why the correlations that we propose with the cultural phases are hypothetical. All along the last two thousand years, a new agricultural phase is recorded at the top of the pollen diagram, by a last phase of abandonment.

Probably during Roman Times (LPAZ 10a), there is a dramatic decline of the forest which concerns both the maquis and the riparian forest. Olive and vine are cultivated, and livestock is reared as indicated by fairly high rates of nitrophilous species.

A peak of chestnut cultivation (LPAZ 10b) is recorded during a period that could be contemporary to the invasions phase, suggesting that instability caused by the invasions did not affect agricultural activities in the lower Taravo Valley. During this period, there is also a very significant collapse of the green oak groves.

A new phase of cultivation of vine and cereals is again recorded at a period that could be contemporary to the Pisan Phase (LPAZ 10b).

During the Genoa Period (LPAZ 11), the agro-pastoral activities decline allowing the forest (mainly the maquis, the riparian forest and the pine forest) to reinstall.

## Rapid landscape changes and consequences for human occupation

During the first 5000 years of the Holocene, the Corsican vegetation is modified mainly according to the climatic changes, with the establishment of a vegetation dominated first by the pine then the installation of a maquis with *Erica arborea* [149]. The human impact is negligible up to around 7613 + 180–140 cal BP, where a strong burning episode is recorded at Lake Creno and in Lake Bastani, resulting in an opening of the *Erica arborea* forest [149]. Between 7200 cal. BP and 6400 cal. BP, the forest with *Quercus ilex* and *Erica arborea* settles. From the Chalcolithic Age (3000 BC), the human pressure increases. Sedimentological and palynological data show a rapid, even brutal, evolution of the landscape since and, in particular, during the Bronze Age (2300–900 BC) (Fig 11). As demonstrated in other areas of the Western Mediterranean, reinforced sedimentation during arid and cold periods centered around 2.2 ka BC, 1.2 ka BC and LIA (1450–1850 AD) events [78, 80] can partly explain the low valley stratigraphy of Corsica. However, we must not neglect the contribution of human activities in these important aggradations.

**The late chalcolithic age (~3000 BC): Relative landscape stability and dense human occupation.** The I Calanchi/Sapar' Alta archaeological site, at the north of the Canniccia Marshes, dates back to Neolithic Age and shows a significant increase in human occupation

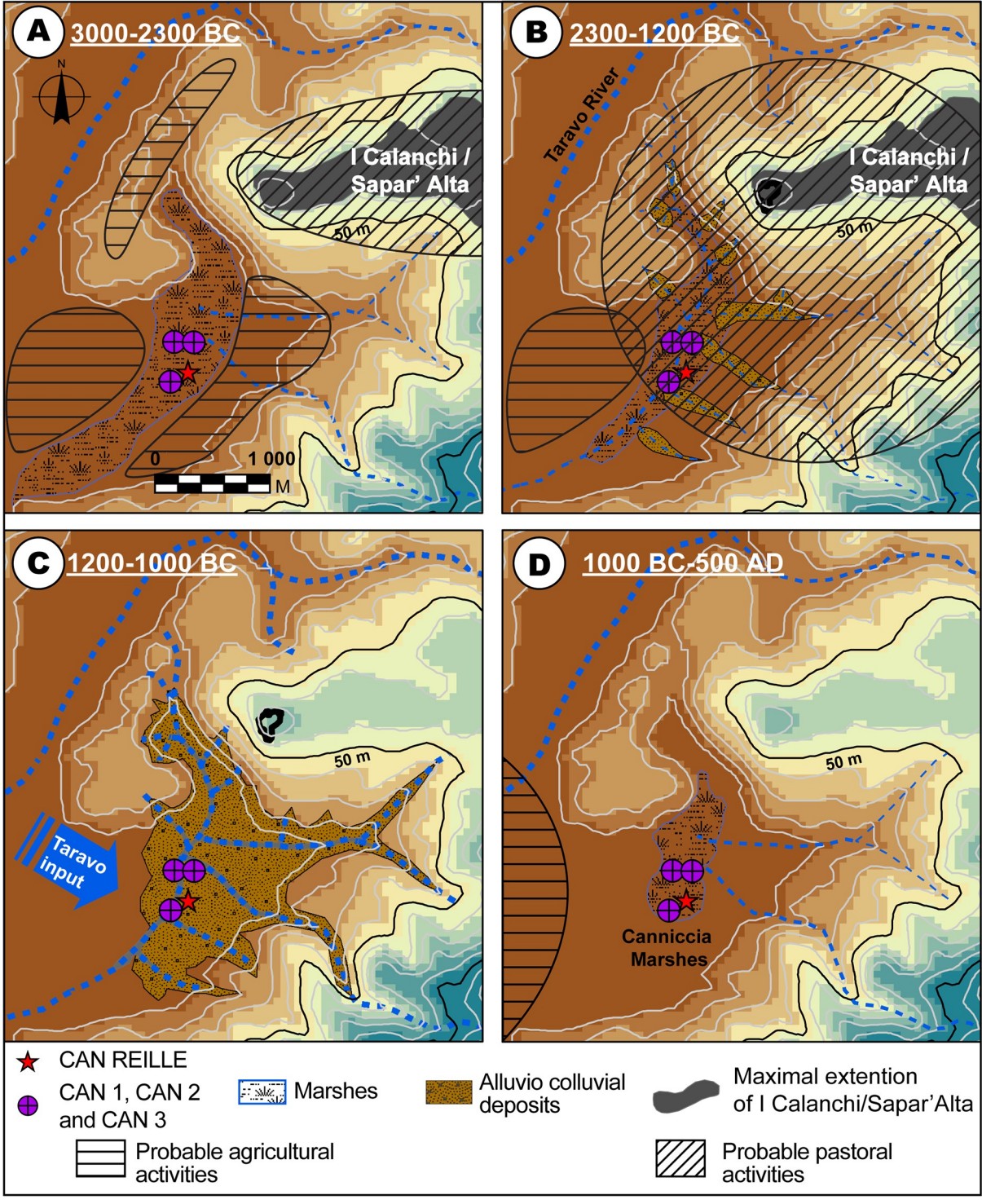

**Fig 11. Palaeogeographic reconstruction of the Canniccia marshes evolution for the last 5000 years. A: ~3000–2200 BC (Late Chalcolithic); B: 2300–1200 BC (Late Chalcolithic to Middle Bronze Age); C: 200–1000 BC (Late to Final Bronze Age); D: 1000 BC– 500 AD (Final Bronze Age to Roman times).** The data for the DEM (®BdAlti25 data) was acquired from the IGN Map Viewer (public domain; ®IGN2019).

during the Chalcolithic Age (3600–2200 BC) (Fig 11A). Terrinian site, of which I Calanchi/ Sapar' Alta is an example, occupy various geographical positions (on plains or rocky ridges) and many of them benefit from dominant heights above favorable ecosystems. Sites co-visibility was important for the surveillance and protection of herds, crops and men [104, 109]. At the end of Middle Chalcolithic Age (~3000 BC), the landscape of the lower Taravo Valley was characterized by a fluvio-torrential event (D1). This period demonstrates also that agricultural activities had little influence on forests that covered the watershed (LPAZ 5). No brutal abandonments are recorded on the sites of the low valley, suggesting that populations still occupied the area or that they moved to other places on the island. The Late Chalcolithic Age (3000– 2300 BC) shows a significant increase in agropastoral activities in the lower Taravo Valley during a period of marshy environments (T1) of 700 years. Agricultural activities are attested on the site of I Calanchi/Sapar' Alta by the discovery of abundant millstones. The high pollen values of olive, and the notations of vine and cereals (LPAZ 6) in the vicinity of the site, confirms that plants were produced locally (Fig 7). Faunal remains from the archaeological site attest to a dominance of sheep and goat whose herds may be at the origin of the decline of the maquis (LPAZ6). A brief episode of agricultural abandonment between 2500 and 2200 (LPAZ 7) is marked by a destabilization of the sediment cover (D2) between 2300 and 1700 BC. D2 can be associated with the 2.2 ka BC event identified in Western Mediterranean [34, 137]. No signs of abandonment are recognized in archaeological excavations and a continuity of occupation may have characterized this period [103], indicating that human populations were able to adapt to climate changes.

**The late chalcolithic to Middle Bronze Ages (2300–1700 BC): Increasing human impact on landscape.** At the transition to the Early Bronze Age (~2000 BC) (Fig 11B), I Calanchi/ Sapar' Alta shows a significant transformation of architectural models and agropastoral activities. It is probably during this period that the archaeological site of I Calanchi/Sapar' Alta developed a monumental defense system composed of circular towers (torre) and fortified enclosures (castelli). The site also bears witness to bronze work, and agropastoral activities are mainly oriented towards pastoralism as indicated by pollen data too (LPAZ 8) (Fig 7). Archaeological excavations show a significant increase in the consumption of beef, coupled in a significant way with pork and goats [114]. This diversification of pastoral practices is accompanied by a decrease in forest cover (between 2250 and 1500 BC) which affects both the slopes and the lower valley of Taravo (LPAZ 8). Increase in plant consumption is also illustrated by a significant number of millstones discovered on I Calanchi/Sapar' Alta [150]. The reduction of olive and cereal in the pollen diagram (Fig 7) during the same period suggests a shift of the production zone towards areas further away from the Canniccia Marshes. The Late Chalcolithic to Middle Bronze Ages correspond to an intense colluvial phase between 2300 and 1700 BC (D2). Vegetation composition and archaeological excavations do not show any notable changes or abandonment during this period. On the other hand, agropastoral, metallurgical and ceramics-related activities may have led to leaching of sediments by exposing the slopes around the Canniccia Marshes. Human activities may thus have played an important role in landscape transformation during the transition from the Early to the Middle Bronze Ages. The beginning of the Late Bronze Age (1350–1200 BC) took place after 500 years of morphosedimentary stability (T2). During this period, human occupation of I Calanchi/Sapar' Alta remained very important and demonstrated the same characteristics as during the Middle Bronze Age. Indeed, the low and mid valley of the Taravo reveals numerous Early to Middle Bronze Age settlements located on rocky promontories. This continuity of occupation (more than 2000 years) demonstrate high human resilience to environmental evolution.

**The late to final Bronze Age (1200–1000 BC): RCC during intense human activities.** During the Late to Final Bronze Age (1300–900 BC), the lower Taravo Valley is marked by

important landscape changes related to significant sociocultural transformations (Fig 11C). Between 1200 and 1000 BC, Canniccia Marsh fillings are composed of alluvial torrential deposits (D3) with particularly important sedimentation rates (75 cm per century). This event can reasonably be related to climatic cooling and increased aridity around 1.2 ka BC as observed in Western Mediterranean [78, 80]. During this period, human populations deserted I Calanchi/Sapar'Alta and other settlements of the lower Taravo Valley [107]. Pollen data (LPAZ 9, Fig 7) shows a significant decrease in cultivated plants which could confirms this hypothesis. It is possible that humans still occupied the lower valley, but agropastoral activities were almost negligible, allowing the forest to reinstall in the valley. The middle sector of the watershed, on the other hand, experienced population growth during the Middle to Late Bronze Age [113]. The geological substratum, relatively more magnetic in the middle valley, constitutes the sedimentary charge of the Taravo. Agropastoral activities may have contributed to soil erosion in this sector and may explain the increased concentration of magnetic grains that we recorded in Canniccia marshes (D3b) during more intense floods.

**From final Bronze Age to Roman Times (1000 BC-450 AD): Landscape stability and progressive re-occupation.** After this period of intense detritism, the Canniccia Marshes reveal a long period of morphosedimentary stabilization (1500 years, T3) and a return to marshy environments (Fig 11D). For nearly 1000 years, the lower Taravo Valley reveals no significant influence from human activities on maquis forest. Between 250 BC and 450 AD, agropastoral impact on vegetation increased progressively. Cultivated plant assemblages characterize the Roman Period [151]. Pastoralism, cereals, olive and vine cultivation are attested to across the island [54] and are well represented in the lower valley of the Taravo. On the other hand, no notable archaeological settlement has been identified in this sector.

**From invasions to Genoan occupation (500–1600 AD): Landscape progressive stabilization and progressive re-occupation.** The Roman Period is followed by a phase of invasions that led to the abandonment of lower valley habitats between 450 and 1100 AD. However, at this period, a new peak of pastoralism and the cultivation of *Castanea* are recorded, indicating the weak influence of invasions on agriculture. Pisan and Genoese occupation (from 1000–1100 AD) is notable in the lower valley of the Taravo Valley. This led to increased anthropogenic pressure characterized by the cultivation of Cerealia, *Vitis* sp., *Castanea sativa* and by pastoralism. At this time, sedimentary filling of the Canniccia Marshes did not allow us to obtain fine resolution of neither the MCA (~750–1000 AD) nor the LIA (1350–1850 AD).

## Conclusion

In this article, we present new paleoenvironmental data that clarify the interrelationships between human societies, vegetation and climate in the Lower Taravo Valley since 3000 BC. Four cores extracted from Canniccia Marshes demonstrate a rapid, even brutal, landscape evolution during the Chalcolithic Age (~3000 BC), in particular the Bronze Age (2000–900 BC). During the Chalcolithic-Terrinian Age, this valley was densely populated and several proto-villages developed in the lower valley. Agricultural activities were well represented, including the growth of cereals, but also vine and olive cultivation. Early domestication may have been of local origin, but could include a diffusion of networks coming from Eastern Mediterranean as well. Pastoralism is attested to notably through sheep and goat herding, but may not have composed the majority of agricultural activities. Archaeological settlements show a certain cultural continuity between the Chalcolithic and Bronze Ages, especially during periods of sedimentary stability and the development of marshy environments. It would seem, therefore, that social-cultural events, undoubtedly combined with important hydro-climatic events (2.2 ka and 1.2 ka BC [34]), led people to adapt their domestic and agropastoral activities. This is reflected

initially in the transition from farming activities to pastoral activities and the construction of defensive structures (castelli) between the Chalcolithic and Early Bronze Ages. Then, between the Middle and Late Bronze Ages, the castelli gradually moved towards the interior of the island [113]. These important transitions unfolded without any decline for the Bronze Age civilizations. Findings indicate a socio-cultural reorganization in the face of a new environmental context rather than a real decline, as research in the Eastern Mediterranean may suggest [21]. Questions on access to and availability of freshwater resources remain to be answered during the Late Bronze Age. Drought in the lower valleys could have pushed some populations to take refuge in the interior of the island, where medium and high mountain lakes provide pure water. Menhir statues along the banks of the Taravo are part of a network present across the island. They may have been erected during the Late Bronze Age (1350–1200 BC) [119–120] in a context of impoverishment of water resources due to a colder and arid climatic context centered on the 1.2 ka BC climatic event. These artifacts, whose symbolism integrates an environmental dimension [120], may thus signify a territorial appropriation of fresh water and indicate the axis of circulation. The adaptation of ancient societies to environmental change is also a question. Mid-mountain areas of Northern Corsica have shown that populations conserved fertile lands during LIA climatic destabilization through terrace construction [44]. This may have also held for the Chalcolithic and Bronze Age periods, although such agricultural structures are not yet identified at this time in the lower Taravo Valley. The adjustment of agropastoral activities, architectural complexity and the spatial reorganization of habitats were the adaptations that Neolithic to Bronze Age societies developed in the face of relatively rapid environmental events. Our results clearly demonstrate that the Bronze Age civilization did not collapse as a result of extreme landscape changes, but that human groups moved to more protected sectors in Corsica. These findings provide evidence of successful socio-cultural developments and highlight human resilience to climate changes.

Further paleoenvironmental research is needed in the Taravo Valley, particularly with regard to the hydro-sedimentary evolution of the Taravo alluvial plain. Shoreline position and costal paleogeography have to be studied as well. These studies will have to focus on the middle part of the watershed where many Late Bronze Age settlements have been found. They will better be able to describe the Chalcolithic to Bronze Age transition in order to identify the societal consequences of climate change. Finally, sedimentary formations close to Roman and Medieval occupation centers will also have to be studied in order to better identify societies interactions with landscape during the MCA and the LIA.

## Supporting information

**S1 Text. Material and methods.**
(DOCX)

**S1 Fig. Unpublished pollen profile CAN REILLE.** Reille 1998.
(TIF)

## Acknowledgments

The authors would like to thank Philippe Blanchemanche, Gaël Piquès from CNRS (UMR 5140, Archéologie des sociétés Méditerranéennes) and Christophe Jorda (INRAP) for providing coring facilities and fieldwork assistance. John Wayne Janusek (Vanderbuilt University, USA) significantly helped to improve the manuscript by polishing the English. Finally, all people who facilitated the fieldwork and provided authorizations are hereby warmly acknowledged.

## Author Contributions

**Funding acquisition:** Marc-Antoine Vella, Joseph Cesari, Matthieu Ghilardi.

**Investigation:** Marc-Antoine Vella, Valérie Andrieu-Ponel, Joseph Cesari, Franck Leandri, Kewin Pêche-Quilichini, Maurice Reille, Yoann Poher, François Demory, Doriane Delanghe.

**Methodology:** Marc-Antoine Vella.

**Project administration:** Marc-Antoine Vella, Valérie Andrieu-Ponel, Joseph Cesari, Franck Leandri, Matthieu Ghilardi, Marie-Madeleine Ottaviani-Spella.

**Supervision:** Marc-Antoine Vella, Valérie Andrieu-Ponel, Joseph Cesari, Franck Leandri, Matthieu Ghilardi, Marie-Madeleine Ottaviani-Spella.

**Validation:** Marc-Antoine Vella.

**Visualization:** Marc-Antoine Vella, Valérie Andrieu-Ponel.

**Writing – original draft:** Marc-Antoine Vella, Valérie Andrieu-Ponel.

**Writing – review & editing:** Marc-Antoine Vella, Valérie Andrieu-Ponel.

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
