## [Decision Letter · Decision Letter 0]

31 Jul 2019

PONE-D-19-17674

Early Impact of Agropastoral Activities and Climate on the Littoral Landscape of Corsica during Mid-Holocene

PLOS ONE

Dear Dr. Vella,

Thank you for submitting your manuscript to PLOS ONE. After careful consideration, we feel that it has merit but does not fully meet PLOS ONE’s publication criteria as it currently stands. Therefore, we invite you to submit a revised version of the manuscript that addresses the points raised during the review process.

We would appreciate receiving your revised manuscript by Sep 14 2019 11:59PM. To enhance the reproducibility of your results, we recommend that if applicable you deposit your laboratory protocols in protocols.io, where a protocol can be assigned its own identifier (DOI) such that it can be cited independently in the future. For instructions see: http://journals.plos.org/plosone/s/submission-guidelines#loc-laboratory-protocols

We look forward to receiving your revised manuscript.

Kind regards,

Andrea Zerboni, Ph.D.

Academic Editor

PLOS ONE

Journal Requirements:

2. We note that Figures 1 and 2 in your submission contain copyrighted images.

All PLOS content is published under the Creative Commons Attribution License (CC BY 4.0), which means that the manuscript, images, and Supporting Information files will be freely available online, and any third party is permitted to access, download, copy, distribute, and use these materials in any way, even commercially, with proper attribution. For more information, see our copyright guidelines: http://journals.plos.org/plosone/s/licenses-and-copyright.

a.         You may seek permission from the original copyright holder of Figures 1 and 2 to publish the content specifically under the CC BY 4.0 license.

3. Please include your tables as part of your main manuscript and remove the individual files.

Please note that supplementary tables should be uploaded as separate "supporting information" files

6. Please ensure that you refer to Figure 8 in your text as, if accepted, production will need this reference to link the reader to the figure.

7. Please include captions for your Supporting Information files at the end of your manuscript, and update any in-text citations to match accordingly. Please see our Supporting Information guidelines for more information: http://journals.plos.org/plosone/s/supporting-information

Additional Editor Comments:

This is a very good paper but reviewers suggest to do a bit more work before we can accept this manuscript. I went through the paper and I think the reviewers highlighted some important points to be fixed. I therefore suggest moderate revisions

Reviewers' comments:

Reviewer's Responses to Questions

**Comments to the Author**

1. Is the manuscript technically sound, and do the data support the conclusions?

Reviewer #1: Yes

Reviewer #2: Yes

2. Has the statistical analysis been performed appropriately and rigorously? 

Reviewer #1: N/A

Reviewer #2: Yes

3. Have the authors made all data underlying the findings in their manuscript fully available?

Reviewer #1: Yes

Reviewer #2: Yes

4. Is the manuscript presented in an intelligible fashion and written in standard English?

Reviewer #1: Yes

Reviewer #2: No

5. Review Comments to the Author

Reviewer #1: Overall comments:

This is an interesting paper that presents a well-argued palaeoenvironmental research in the Taravo Valley, a littoral plain of the southwest coast of Corsica rich in archaeological evidence beginning in the Chalcolithic Age. The multidisciplinary approach (sedimentology and geomorphology combined with palynology) is deeply developed to reconstruct the landscape transformations of this area and infer the interrelationships between humans, climate and vegetation during the middle and late Holocene. The multiproxy data (on sediments and pollen) are well integrated and discussed, and provide detailed information on landscape dynamics and past land-uses.

The manuscript is well designed and consistent, although I suggest few changes to facilitate its reading and comprehension before being published.

Main comments:

1) As the paper focuses on the early impact of human activities and climate on the landscape during the mid-Holocene, it would be useful to briefly describe the environmental setting/plant landscape shortly before the stable human occupation of the area during the Chalcolithic Age. For this purpose, the Authors could add/compare in the discussion the unpublished data of other (few) samples from CAN REILLE prior to 3000 BC (S1 Figure). The Authors may wish to consider this comparison in order to investigate the vegetation changes at the passage from wild to human environments (see for example: Mercuri et al. (2013) Quaternary International 303: 24-42).

2) Regarding the methodological section, I suggest the Authors specify the main plant taxa indicators of agropastoral activities; except for the well-known crop species, pasture plants are not so commonly known (in particular, for readers not familiar with botany, clarify the ‘pastoral’ meaning of the nitrophilous plants listed in the caption of Fig. 7). 3) Please, reconsider the timing of the most ancient testimonies of vine cultivation in Italy (L. 455-456): the results of the research by Ucchesu et al. (2015; Vegetation History and Archaeobotany 24: 587-600) provide evidence that cultivated grapevines were present in Sardinia during the Late Bronze Age, also suggesting that the selection and domestication processes of the wild grape may have started during the Middle BA.

Minor comments:

1) Chapter 2.2: please provide the author(s) for each plant name at the species level.

2) Chapter 2.5: the title should be modified as in this section (and throughout the whole text) there is any specific reference to the Mesolithic Age.

2) Chapter 5.1: the sub-heading titles ‘Four detritic phases…’ (L. 328), but below are presented five detritic events (L. 330); in addition, the list of these phases lacks the point 4 (I guess ‘4)’ should be before ‘D4’ at L. 339), and the point 5 refers to the opposite bank of the Taravo. The different steps of detritism should be clarified.

3) L. 407-408: The sentence is unclear: with ‘15 fires’ do you mean 15 big fire events? I could be wrong, but in the reference you quoted I didn’t see any information about ‘number of fires’; Carcaillet et al. deals with the role of fire in the past vegetation changes and confirm an increase of fire frequency since 2500 BP probably driven by the slash-and-burn agriculture. In addition, the period 2500-2000 BC is not 1000 years. Please rephrase.

4) Please, check for the accuracy of some words (e.g., L. 147-149: ‘maquis’, ‘pine forests’, ‘subsp.’ not in italics; L. 246-252: the abbreviations of the authors of plant names are not in italics, and ‘Sylvestris’ not capitalized; L. 436-437: ‘dicoccum’ instead of ‘dicoccun’, ‘Vicia ssp.’ instead of ‘Vivia ssp.’, and ‘fabaceae’ with the first letter capitalized), numbers/dates (e.g., L. 212: add the dash between 1400 and 700, and change ‘BC’ to ‘AD’; L. 631: ‘1350’ instead of ‘135000’) and graphic symbols (e.g., L. 332: comma instead of bracket after 3000 BC; L. 367 and L. 369: change ‘>’ to ‘<’ and vice-versa).

5) Table 2: the abbreviations of the authors of plant names are not in italics; dicoccum’ instead of ‘dicoccun’, and ‘Sylvestris’ not capitalized.

6) Table 5: in LPAZ 10, ‘Olea’ should be in italics.

7) Fig.1 caption: please specify that b. Bastani, c. Creno, etc… are reference sites.

Reviewer #2: General considerations.

The manuscript entitled “Early impacts of agropastoral activities and climate on the littoral landscape of Corsica during Mid-Holocene” aims to documents the Mid to Late Holocene landscape history of the lower Taravo valley in Corsica. It provides valuable new knowledge about alluvial geomorphological evolution and vegetation dynamics under climatic and human pressure, which enriched previous studies in the lowland and coastal areas of Corsica. The multidisciplinarity approach in sedimentology and palynology meet the standards of the disciplinary fields, together with a rich state of the art of local archaeological knowledge. Finally, this study offers the opportunity to publish a part of an original unpublished pollen analysis from 1998.

My main hesitations relate to the manuscript itself, which I feel needs considerable improvement to promote the results. Firstly, the title is unclear: it introduces “Early impact” and “Mid-Holocene period”, while the paper considers a period from 5000 BP (3000 BC) to present day, namely from the transition between Mid-Holocene and Late Holocene and during the whole Late Holocene. Then, much would be gained with a more clearly exposed aim of the paper. Many approximate sentences make the manuscript not as accurate as it might be. Here is an example: p. 8, l.349, the authors wrote “A detrital event D1 (around 3000 BC) is not recorded as major climatic events in the Western Mediterranean”. Wouldn’t it be more accurate to write “The detrital event D1 (around 3000 BC) in the Cannicia marshes does not correspond to any major climatic events recorded in the Western Mediterranean”? Other kind of imprecisions with suggestions are detailed below (Specific comments). Period names are not standardized. The term “Late Holocene” seems more commonly used than the “recent Holocene Epoch” the authors choose at the beginning of the text. Thereafter, “Upper Holocene” is found as well. In the first discussion paragraph (5.1), it would be clearer to describe the Detritism event (D) in the same way than the Phases of Morphosedimentary stability (T).

Obviously, paragraph 5.3 is in a much more completed form than the other part of the text, both for argumentation and English expression (I encourage the authors to have their manuscript proofread to improve the English expression). All the other parts of the manuscript should reach this quality to achieve the goal of the publication in PlosOne.

Specific comments (by page/line numbers):

p.2 l. 56 to 58: The references cited by the authors did not attest to the presence of RCC (identified and published by Mayewski et al. 2004 at the hemispheric scale), but they highlight or document the regional/local effect of these RCC.

l. 72-73: remove “recent Holocene” if the considered period is …” since the Last Glacial Maximum”

l. 84: Please explain why “archaeological research programs”… “required more paleoenvironemental analysis”.

l. 86: “(“ missed

p.3 l. 147 to 148: “maquis” and “pine forest” might be regular not italic.

p.4 l. 135: “… along with maquis such as…”, add species after maquis.

p. 4/5. The paragraph 2.4 would be more relevant and useful for the reader with a critical review of the available data: date of the analysis, location, chrono-stratigraphical control of the sequences… (ref. [55] Reille et al. 1997, is about Late Glacial). A chronological phases description would be more appropriate than the somewhat “oldfashion” chronozones.

p.5 l. 212: remove “1300-1400”

p.6 l. 292: Please precise which application of Blaauw you used (Clam, I think so?) and which version. Idem on Fig. 3

l. 292-295: This sentence is unclear for me. What are the dates between brackets, the earliest radiocarbon dating for each sequences?

p.7 and Tab. 5: it would be nice to provide here the chrono-stratigraphic boundaries of the LPZ, and to precise the way you called the LPAZ. It is strange to begin a sequence with a subzone 5c.

On table 5: remove “7:” and “10:”

p.7 l.323: You talk about “four cores” that “highlight four phases of intense detritism”. Didn’t you studied the sediments of only CAN 1 to 3?

l.326: Isn’t it Fig. 8 instead of Fig. 9?

l.332: What is: “. 2200 BC)”?

l.339: “4)” is missing

p. 8 l.351-352: more accurately, “An event that was coeval with D2 is also observed…..”

l.378: What is D5?...

p. 9 l.407-408: please be more precise, the fire history derives from the Creno sequence study.

l.409: “It attests to the intensity of Chalcolithic and Early Bronze Age agropastoral activities on vegetation”. This rather might be an argued assumption.

l.422/432: the references about Olea europaea in the Mediterranean are too old, and then the dating is not up to date. Please see for example: Carrion et al. QSR 29, 2010; Breton et al. C.R. Biologie332, 2009 or Newton et al. VHA, 2013.

l.438: A diversity of agricultural…

p.10 l.472: the decrease of arboreal pollen (instead of the frequencies)

l.488: beginning the sentence with “During Roman Times…” is not consistent with what you reminded just above about “the age model of the last 2000 years is weak…. the correlations that we propose with the cultural phases are hypothetical…”

p.12 l.567-572: the authors assume that, in the Lower Taravo Valley, “important landscape changes … may have precipitated significant sociocultural transformations”. What about the opposite? Isn’t it possible to also discuss the hypothesis that “significant sociocultural transformations have precipitate important landscape changes”. What is in good accordance with the last sentence of the paragraph “Agropastoral activities may have contributed to soil erosion…”

p. 13 l.601: Cerealia instead of cerealia

References

p. 22 l. 1047: 1961 (6 is missing)

Tables and Figures:

Tab.3 The bibliographical reference is wrong: [120] instead of [122].

Fig. 7 The “Nitrophilous plants” curve actually sums all the apophytes: nitrophilous and ruderals. I do not understand why some taxa that are in this “Nitrophilous plants” curve, such as Asteroideae sum of Rumex, are displayed alone as well. It makes the diagram unclear.

Fig. 10 Nice synthetic figure. Just two remarks: 1) samples without pollen have got values, despite the white background, 2) the first “Occupation phases” graph, just after “Nitrophilous plants” curve, does not have an header (Corsica ?).

Fig. 11 A and B: text in black on dark grey is unreadable

6. PLOS authors have the option to publish the peer review history of their article (what does this mean?). If published, this will include your full peer review and any attached files.

Reviewer #1: No

Reviewer #2: No

---

## [Author Response · Author response to Decision Letter 0]

31 Oct 2019

Dear Dr Zerboni, academic Editor, and dear anonymous reviewers, 

I would like to thank you personally for your attentive reading of the article that my research team and I have proposed for publication in Plos One. I also thank you for your interest in this innovative multidisciplinary approach on past societies/environments interactions. We have taken into account all your observations, corrections and addition of bibliographical references. The general and more specific remarks you have given us allow us to significantly improve the quality of our article. We also hope that our work will make a significant contribution to the knowledge of the Mediterranean environments.

Before responding point by point to each of the reviewers, we would like to respond more generally to your comments. As you have rightly pointed out, we have sought to clarify certain points in our manuscript. 

 The chrono-cultural context of our study has been clarified. First in the title highlighting that the mid Holocene constitutes the upper temporal limit. Then in the text, changing the title of some parts but also adding the chronological boundaries of events identified by palynology and geomorphology. 

 The phases of detritism have been rectified to better correspond to the specific environment of the Canniccia pond (D1 to D4 and not D5). In the same way the presentation of the different environments (D and T) has been homogenized in order to simplify the understanding.

 The missing bibliographical references that you pointed out to us were of great help for our study and allowed us to refine our chronology of the domestication phases of the olive and the vine. They also allowed us to better understand island cultural processes but also to include Corsica in a more regional, Mediterranean-wide context.

 Finally, we made a last proofread to improve the English expression

Here is, in the following pages, the answers to the corrections made in the manuscript.

Thank you very much for your participation to our work, I hope to meet you at an international event around the Mediterranean Worlds.

Best regards

Marc-Antoine Vella

---

## [Decision Letter · Decision Letter 1]

26 Nov 2019

Early Impact of Agropastoral Activities and Climate on the Littoral Landscape of Corsica since Mid-Holocene

PONE-D-19-17674R1

Dear Dr. Vella,

We are pleased to inform you that your manuscript has been judged scientifically suitable for publication and will be formally accepted for publication once it complies with all outstanding technical requirements.

With kind regards,

Andrea Zerboni, Ph.D.

Academic Editor

PLOS ONE

Additional Editor Comments (optional):

Reviewers' comments:

Reviewer's Responses to Questions

**Comments to the Author**

1. If the authors have adequately addressed your comments raised in a previous round of review and you feel that this manuscript is now acceptable for publication, you may indicate that here to bypass the “Comments to the Author” section, enter your conflict of interest statement in the “Confidential to Editor” section, and submit your "Accept" recommendation.

Reviewer #2: All comments have been addressed

2. Is the manuscript technically sound, and do the data support the conclusions?

Reviewer #2: Yes

3. Has the statistical analysis been performed appropriately and rigorously? 

Reviewer #2: Yes

4. Have the authors made all data underlying the findings in their manuscript fully available?

Reviewer #2: Yes

5. Is the manuscript presented in an intelligible fashion and written in standard English?

Reviewer #2: Yes

6. Review Comments to the Author

Reviewer #2: (No Response)

7. PLOS authors have the option to publish the peer review history of their article (what does this mean?). If published, this will include your full peer review and any attached files.

Reviewer #2: No

---

## [Editor Report · Acceptance letter]

5 Dec 2019

PONE-D-19-17674R1 

Early Impact of Agropastoral Activities and Climate on the Littoral Landscape of Corsica since Mid-Holocene 

Dear Dr. Vella:

I am pleased to inform you that your manuscript has been deemed suitable for publication in PLOS ONE. Congratulations! Your manuscript is now with our production department. 

With kind regards,

on behalf of

Prof. Andrea Zerboni 

Academic Editor

PLOS ONE